# MMAD: A Comprehensive Benchmark for Multimodal Large Language Models in Industrial Anomaly Detection

**Xi Jiang**[1]    **Jian Li**[2]    **Hanqiu Deng**[3]    **Yong Liu**[2]    **Bin-Bin Gao**[2]    **Yifeng Zhou**[2]
**Jialin Li**[2]    **Chengjie Wang**[2,4]    **Feng Zheng**[1]*
[1]Southern University of Science and Technology    [2]Tencent YouTu Lab
[3]University of Alberta    [4]Shanghai Jiao Tong University
`jiangx2020@mail.sustech.edu.cn, hanqiu1@ualberta.ca,`
`{swordli, choasliu, danylgao, joefzhou, jarenli, jasoncjwang}`
`@tencent.com, f.zheng@ieee.org`

## Abstract

In the field of industrial inspection, Multimodal Large Language Models (MLLMs) have a high potential to renew the paradigms in practical applications due to their robust language capabilities and generalization abilities. However, despite their impressive problem-solving skills in many domains, MLLMs' ability in industrial anomaly detection has not been systematically studied. To bridge this gap, we present **MMAD**, a full-spectrum **MLLM** benchmark in industrial **A**nomaly **D**etection. We defined seven key subtasks of MLLMs in industrial inspection and designed a novel pipeline to generate the MMAD dataset with 39,672 questions for 8,366 industrial images. With MMAD, we have conducted a comprehensive, quantitative evaluation of various state-of-the-art MLLMs. The commercial models performed the best, with the average accuracy of GPT-4o models reaching 74.9%. However, this result falls far short of industrial requirements. Our analysis reveals that current MLLMs still have significant room for improvement in answering questions related to industrial anomalies and defects. We further explore two training-free performance enhancement strategies to help models improve in industrial scenarios, highlighting their promising potential for future research. The code and data are available at `https://github.com/jam-cc/MMAD`.

## 1 Introduction

Automatic vision inspection is a crucial challenge in realizing an unmanned factory (Benbarrad et al., 2021). Traditional AI research for automatic vision inspection, such as industrial anomaly detection (IAD) (Jiang et al., 2022b; Ren et al., 2022; Zhang et al., 2024b), typically relies on discriminative models within the conventional deep learning paradigm. These models can only perform trained detection tasks and cannot provide detailed reports like quality inspection workers. Additionally, when production lines or requirements change, traditional methods necessitate retraining or redevelopment. The development of MLLMs (Jin et al., 2024) has the potential to alter this situation. These generative models can flexibly produce the required textual output based on input language and visual prompts, allowing us to guide the model using language similar to instructing humans.

Nowadays, multimodal large language models, represented by GPT-4 (Achiam et al., 2023), can already do many human jobs, especially high-paying intellectual jobs like programmers, writers, and data analysts (Eloundou et al., 2023). In comparison, the work of quality inspectors is simple, typically not requiring a high level of education but relying heavily on work experience. Therefore, we are greatly interested in the question:

*How well are current MLLMs performing as industrial quality inspectors?*

---

*Correspondence to Feng Zheng (f.zheng@ieee.org)

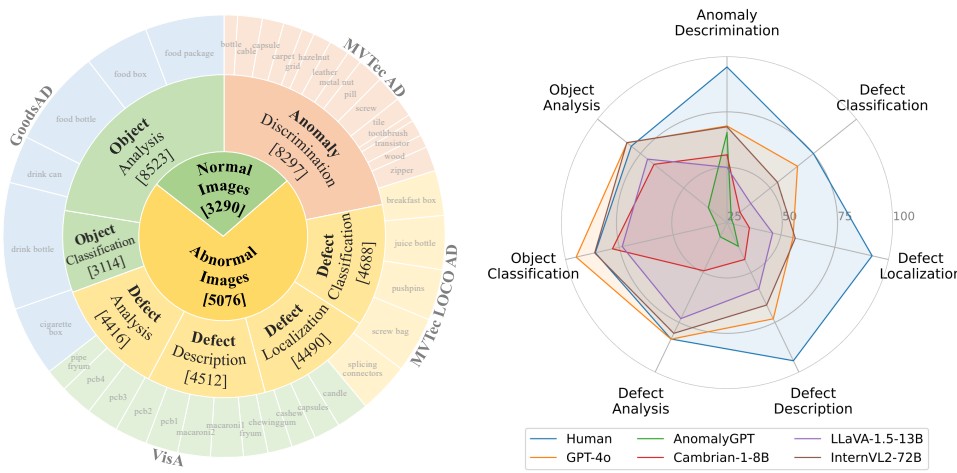

Figure 1: Left: Innermost layer: image components, middle layer: subtasks composition, outermost layer: object categories. MMAD covers 7 key subtasks and 38 representative categories of IAD. Right: Results of 5 representative MLLMs and Human. The left-skewness indicates that models perform well on object-related questions but poorly on questions related to defects.

Recent studies have attempted to explore this issue. Through instruction following mechanisms (Dai et al., 2023; Liu et al., 2024b), some reports evaluate IAD examples with MLLMs, demonstrating the advantages of MLLMs in generalization and flexibility (Zhang et al., 2023a; Cao et al., 2023; Xu et al., 2024). Unfortunately, they only tested a few qualitative examples with no quantitative results. Other studies, such as AnomalyGPT (Gu et al., 2024) and Myriad (Li et al., 2023), specifically trained MLLMs to understand the outputs of traditional IAD models. However, they use the traditional evaluation, which measures the ability of expert models rather than MLLMs. The lack of unified test data and output format also led to an incomplete comparison with other general models. So, a comprehensive benchmark to compare the quantitative results of MLLMs in IAD is necessary.

In this paper, we introduce the first-ever benchmark for MLLMs in IAD tasks. Current MLLMs are challenging to evaluate directly on IAD tasks because existing publicly available datasets only contain visual perception annotations and category labels, lacking rich semantic annotations. To address this issue, we designed a comprehensive pipeline. First, we utilized GPT-4V to generate rich semantic annotations with visual annotations and language interaction. Based on the semantic annotations, we generated questions and options for testing, which were manually reviewed to ensure their reasonableness and accuracy. Ultimately, we collected 8,366 samples from 38 classes of industrial products across 4 public datasets, generating a total of 39,672 multiple-choice questions in 7 key subtasks, as illustrated in the left panel of Figure 1. Our benchmark encompasses representative image data and language-based evaluation methods, providing a fair and reasonable assessment of MLLMs' performance in IAD. Figure 2 provides some examples.

With MMAD, we have conducted a comprehensive, quantitative evaluation of various state-of-the-art (SOTA) MLLMs, including the GPT-4 series and Gemini 1.5 series (Reid et al., 2024), as well as open-source image models like InternVL2 (Chen et al., 2023) and LLaVA-NeXT (Liu et al., 2024a), and industry anomaly detection models like AnomalyGPT (Gu et al., 2024). As shown in the right panel of Figure 1, our experiments demonstrate that GPT-4o is the best-performing model, reaching 74.9%, but as the scale of models increases, some open-source models are gradually approaching the capabilities of these commercial models. However, industrial scenarios often demand high accuracy, and the current accuracy levels are far from meeting practical standards. Additionally, both open-source and proprietary models perform significantly worse when answering questions related to anomalies and defects compared to questions about objects, which reveals an important capability weakness of MLLMs. Notably, the model specifically designed for IAD, AnomalyGPT (Gu et al., 2024), performed the worst overall, as it could only handle anomaly discrimination tasks it was trained on and failed to respond adequately to questions in other subtasks. This indicates that enhancing MLLMs' capabilities in IAD through training requires larger-scale industrial data and the provision of more comprehensive industrial knowledge.

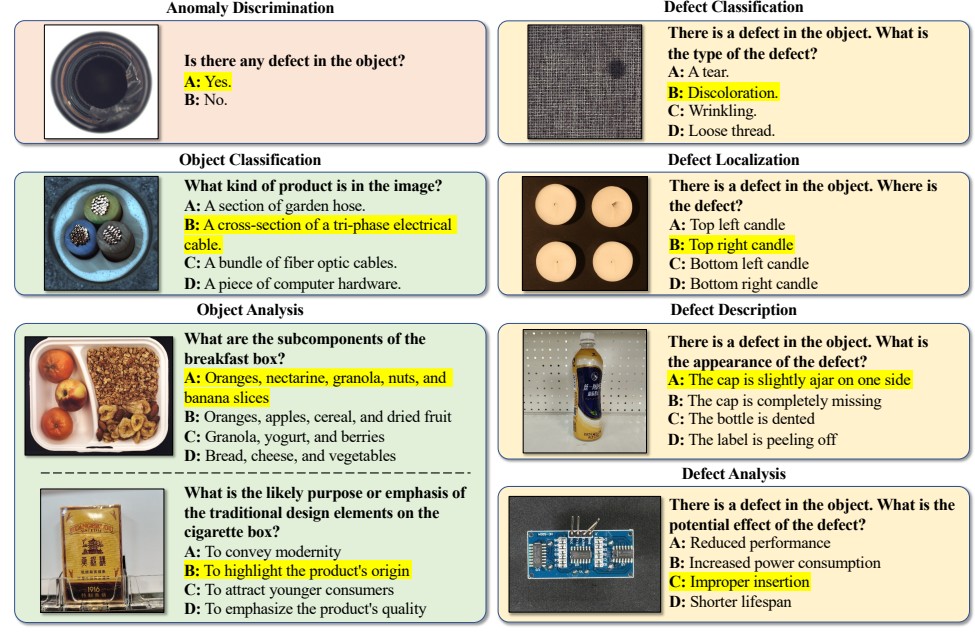

Figure 2: Examples of 7 subtasks of MMAD. Each question is presented in a multiple-choice format and includes several distractor options. We present different categories of objects in various examples to demonstrate the diversity.

Beyond standard testing, we also conducted experiments on different settings and scaling laws, which aid in further comparative studies. Given the current shortcomings of MLLMs in addressing anomalies and defects, we explored two performance boost schemes that do not require additional training: Retrieval-Augmented Generation (RAG) and expert agent, the previous one through text augmentation and the other through visual augmentation. These methods can improve MLLMs' performance in IAD to some extent, but they are still limited by the fundamental capabilities of the models. Overall, current MLLMs are not yet capable of effectively performing the duties of a quality inspector. They require further supplementation with IAD knowledge and enhanced capabilities for fine-grained understanding and cross-comparison of multiple images.

The contributions of this paper are summarized as follows:

- To the best of our knowledge, our proposed MMAD is the first evaluation benchmark for MLLMs in the task of IAD. This benchmark fills the gap in the application of MLLMs in the industrial domain and sets new challenges for their capabilities.

- We introduce a novel pipeline for generating semantic annotations for visual anomaly detection data, addressing the issue that MLLMs cannot be directly evaluated on IAD datasets.

- We comprehensively evaluate the performance of representative MLLMs on MMAD, highlight the weaknesses of current MLLMs in fine-grained industrial knowledge and multi-image understanding, and provide two general boost schemes.

# 2 RELATED WORK

## 2.1 MULTIMODAL BENCHMARKS

In recent years, substantial efforts on benchmarks have been dedicated to exploring MLLMs (Li & Lu, 2024) from various perspectives. However, the capabilities assessed by these benchmarks differ significantly from those required for IAD. Firstly, most MLLM benchmarks primarily focus on single-image input scenarios, such as MME (Fu et al., 2023), MMBench (Liu et al., 2023b), and MMVP (Tong et al., 2024b). In contrast, industrial quality inspection necessitates the ability to process multiple images, as industrial knowledge is highly specialized and often requires additional

Table 1: Statistics on the composition and quantity of MMAD data.

| Image Source | Specialty | Sampled Images | Generated Questions | Object Categories | Defect Categories |
|---|---|---|---|---|---|
| MVTec AD (Bergmann et al., 2019) | A variety of objects and textures | 1691 | 8338 | 15 | 73 |
| MVTec LOCO AD (Bergmann et al., 2022) | Including both structural and logical anomalies | 1566 | 7624 | 5 | 89 |
| VisA (Zou et al., 2022) | Containing multiple instances and complex structure | 2141 | 10622 | 12 | 67 |
| GoodsAD (Zhang et al., 2024a) | Multiple goods in each category | 2968 | 13088 | 6 | 15 |
| SUM | - | 8366 | 39672 | 38 | 244 |

images for product recognition. Secondly, most benchmarks emphasize the general capabilities of MLLMs rather than the specific needs of particular domains. For instance, TextVQA (Singh et al., 2019) focuses on text recognition in images, MATH-Vision (Wang et al., 2024c) and MATH-VISTA(Lu et al., 2023) evaluates mathematical reasoning in visual contexts, and Video-MME (Fu et al., 2024) emphasizes video understanding. Most importantly, these benchmarks do not address the industrial domain. For example, MMMU (Yue et al., 2024) covers lots of fields, such as Art, Business, Science, Social Science, and Engineering, but not industry. Seed-Bench (Li et al., 2024a) and CompBench (Kil et al., 2024) test multi-image comparison capabilities, but both involve fictional tasks and cannot provide realistic references. In contrast, the medical field, which has tasks and data formats similar to the industrial domain, already has numerous benchmarks for MLLMs, including Asclepius (Wang et al., 2024d), GMAI-MMBench (Chen et al., 2024b) and PMC-VQA (Zhang et al., 2023b). Therefore, proposing the first MLLMs benchmark for the industrial domain would fill a significant gap.

## 2.2 INDUSTRIAL ANOMALY DETECTION

Traditional IAD research primarily aims to address a significant issue in industrial visual inspection: discriminating and localizing defects without samples of defects. Consequently, traditional IAD methods typically involve training on a large number of normal samples and then using outlier detection techniques to identify anomalies in test samples. Common approaches include memory bank-based methods (Jiang et al., 2022a; Wang et al., 2025), reconstruction-based methods (Deng & Li, 2022; Jiang et al., 2024a), and methods based on training with synthetic anomalies (Zavrtanik et al., 2021). Recent research has begun to focus on the generalization capability of IAD. Leveraging vision-language models such as CLIP, some few-shot, and even zero-shot models have emerged, such as AnomalyCLIP (Zhou et al., 2023), M3DM-NR (Wang et al., 2024a) and InCTRL (Zhu & Pang, 2024). However, these discriminative models heavily rely on predefined anomaly concepts in the CLIP model, limiting their ability to generalize to new scenarios. For instance, models trained on structural anomalies struggle to detect logical anomalies (Bergmann et al., 2022). The emergence of MLLMs may help address this challenge, as they can understand complex textual inputs and provide diverse responses in conjunction with visual components.

Some studies have already demonstrated the advantages of open input-output formats of MLLMs in IAD. For example, LogiCode (Zhang et al., 2024c) uses MLLMs for logical reasoning and automatically generates code to address different types of logical anomalies. Xu et al. (2024) proposes that designing visual and language prompts can directly enhance the performance of MLLMs in IAD tasks. Additionally, some research has begun to train MLLMs directly on IAD datasets, such as AnomalyGPT (Gu et al., 2024), Myriad (Li et al., 2023), and FabGPT (Jiang et al., 2024b). However, these three models heavily depend on the capabilities of expert models and cannot freely extend their capabilities by changing inputs. Moreover, we have found that due to the small amount of IAD data (Wang et al., 2024b), MLLMs like AnomalyGPT are highly prone to overfitting. In conclusion, current IAD methods rarely meet our paradigm, so we focus our evaluation on general MLLMs.

## 3 THE MMAD DATASET

### 3.1 DATA COLLECTIONS

An excellent industrial quality inspector should be able to adapt to the inspection tasks of different products, as the skills involved in visual inspection are similar. Therefore, our designed benchmark

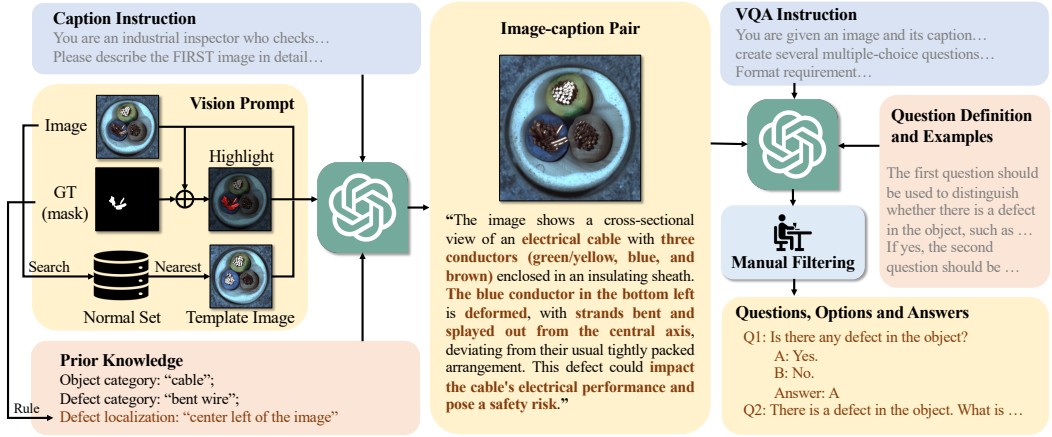

Figure 3: The VQA data generation pipeline for IAD. We utilize images from the open-source IAD dataset and leverage GPT-4V to automate the generation of question-answer texts. Initially, the model is prompted to provide detailed captions for IAD images by summarizing visual cues and textual prior knowledge. Based on these image-caption pairs, the model then generates questions across different subtasks according to predefined question definitions and examples, simultaneously creating multiple-choice questions with several distractor options. Finally, manual verification is conducted to filter out low-quality VQA pairs, resulting in high-quality VQA data for the IAD.

needs to cover multiple scenarios of IAD. We achieved data diversity by collecting and sampling from four different IAD datasets with distinct focuses, resulting in over 38 product categories and 244 defect types, as detailed in Table 1. Among these, MVTec AD (Bergmann et al., 2019) is one of the most prominent datasets for IAD, encompassing multiple categories of objects and textures, where we use finer annotations from Defect Spectrum (Yang et al., 2023). MVTec LOCO AD (Bergmann et al., 2022) focuses on logical anomalies, thereby testing the model's understanding of logical-level anomalies. The VisA (Zou et al., 2022) dataset includes multiple instances and complex examples, reflecting more intricate IAD scenarios. The GoodsAD (Zhang et al., 2024a) dataset primarily consists of industrial goods, and due to the varying appearances of finished products from different brands, it significantly expands the number of object categories.

## 3.2 QUESTION DEFINITION

To evaluate whether MLLMs can fulfill the role of an industrial quality inspector, it is necessary to test a wide range of capabilities. On the production line, a quality inspector must not only identify defective samples but also classify and grade defects, analyze causes, and diagnose faults. Therefore, in addition to basic anomaly detection, we have designed four anomaly-related subtasks and two object-related subtasks, as detailed below:

- **Anomaly Discrimination/Detection**: Asking whether a sample has defects. These questions are binary classification problems that test the ability to judge anomalies in samples.
- **Defect Classification**: Asking about the type of defect. It verifies the model's basis for anomaly discrimination while also testing its understanding of industrial defect categories.
- **Defect Localization**: The model needs to specify the exact location of the defect, further verifying the model's basis for anomaly detection. Since most MLLMs cannot directly output masks, we standardize the testing by using textual descriptions to indicate the location.
- **Defect Description**: Describing the appearance characteristics of the defect. It simulates the steps of collecting information on the size, color, and other attributes of defects in actual production. This information can be used to determine the defect's grade according to production standards and to infer whether the defect is related to equipment malfunction.
- **Defect Analysis**: This involves analyzing the potential impact of the defect on the product, which is used to further determine the defect's severity level.
- **Object Classification**: Categorization of industrial products. An understanding of industrial products aids in recognizing normal characteristics and identifying anomalies.

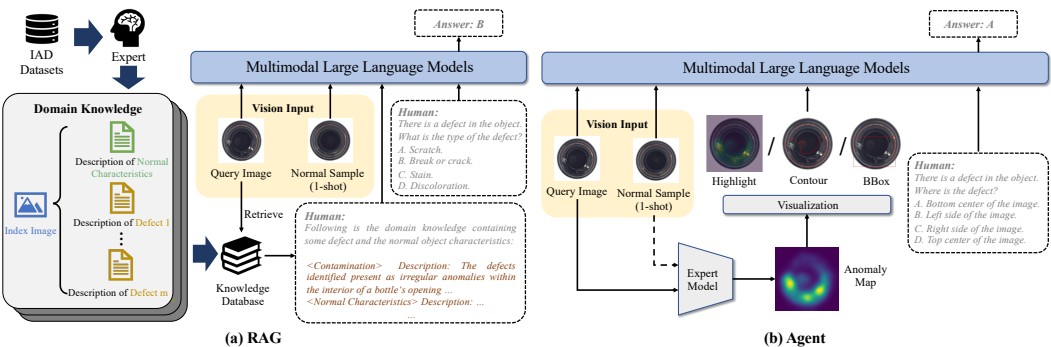

Figure 4: Illustration of two proposed boost methods of MLLMs in MMAD.

- **Object Analysis**: This involves questioning the composition, position, appearance, and function of the object, aiming to assess the detailed understanding of the specific product.

Some examples are shown in the Figure 2. To better evaluate the outputs of MLLMs, we use multiple-choice questions, a method proven effective in previous research (Li et al., 2024a; Liu et al., 2023b). To avoid biases inherent in language models, we randomize the options in our questions and use text matching to select the closest option when the model cannot correctly follow instructions and output the answer's letter.

## 3.3 DATA GENERATION

Due to the lack of semantic annotation in open-source IAD datasets, the currently collected data cannot be directly used to evaluate MLLMs. Therefore, we designed a novel pipeline to generate evaluation questions for each IAD image. As shown in Figure 3, our process leverages the text generation capabilities of GPT-4V (Achiam et al., 2023), combined with rule-based program outputs, language prompts, and human filtering to ensure the reliability of the generated content. First, we generate rich textual descriptions for each image. Since GPT-4V is primarily trained on natural scenes and may struggle with industrial quality inspection, we provide additional visual and textual prompts. In the visual prompts, we highlight the ground truth mask in red on the original image to make the model aware of the defects. Additionally, we retrieve the most similar normal image to serve as a comparison template, using a similarity metric combining the SSIM score (Wang et al., 2004) and the Bhattacharyya distance (Bhattacharyya, 1943) of color histograms. The language prompts include object and defect category labels and textual descriptions of defect positions within the image, utilizing the nine-grid division proposed by Gu et al. (2024). To ensure diversity, we designed instructions for caption generation with variations in the prior knowledge provided, preventing the captions from merely copying the supplied text. Once we have the captions for each image, we generate questions, options, and answers based on predefined subtasks. Unlike natural images, each IAD image corresponds to multiple questions, which are generated simultaneously and then filtered through manual verification. This verification process involved 26 individuals and took over 200 working hours.

## 3.4 BOOST METHODS

**Retrieval-Augmented Generation (RAG).** RAG is a method that combines information retrieval and generation to enhance the performance of language models, particularly for tasks requiring external knowledge (Zhao et al., 2024). Knowledge related to IAD is often specialized and rarely encountered during the training of Multimodal Language Models (MLLMs). Therefore, we propose a RAG method tailored for IAD, as illustrated in Figure 4(a). Experts, with the assistance of large language models, first summarize the existing IAD datasets. For each category, they summarize the characteristics of normal samples and the features of each possible anomaly. The domain knowledge summarized from all datasets forms a retrieval database. During testing, the query image is used to retrieve the relevant category knowledge, which is then incorporated into the text prompts.

Table 2: Performance comparison of both proprietary and open-source MLLMs in MMAD with the standard 1-shot setting. Anomaly Discrimination uses the average accuracy of normal and abnormal categories. (*For the methods not supporting multi-image input, the 0-shot result is reported.)

| Model | Scale | Anomaly Discrimination | Defect Classification | Defect Localization | Defect Description | Defect Analysis | Object Classification | Object Analysis | Average |
|---|---|---|---|---|---|---|---|---|---|
| Random Chance | - | 50.00 | 25.00 | 25.00 | 25.00 | 25.00 | 25.00 | 25.00 | 28.57 |
| Human (expert) | - | 95.24 | 75.00 | 92.31 | 83.33 | 94.20 | 86.11 | 80.37 | 86.65 |
| Human (ordinary) | - | 86.90 | 66.25 | 85.58 | 71.25 | 81.52 | 89.58 | 69.72 | 78.69 |
| Claude-3.5-sonnet | - | 60.14 | 60.14 | 48.81 | 67.13 | 79.11 | 85.19 | 79.83 | 68.36 |
| Gemini-1.5-flash | - | 58.58 | 54.70 | 49.10 | 66.53 | 82.24 | 91.47 | 79.71 | 68.90 |
| Gemini-1.5-pro | - | **68.63** | 60.12 | **58.56** | 70.38 | 82.46 | 89.20 | 82.25 | 73.09 |
| GPT-4o-mini | - | 64.33 | 48.58 | 38.75 | 63.68 | 80.40 | 88.56 | 79.74 | 66.29 |
| GPT-4o | - | **68.63** | **65.80** | 55.62 | **73.21** | **83.41** | **94.98** | **82.80** | **74.92** |
| AnomalyGPT | 7B | 65.57 | 27.49 | 27.97 | 36.86 | 32.11 | 29.84 | 35.82 | 36.52 |
| Qwen-VL-Chat | 7B | 53.65 | 31.33 | 28.62 | 41.66 | 63.99 | 74.46 | 67.94 | 51.66 |
| LLaVA-1.5 | 7B | 51.33 | 37.04 | 36.62 | 50.60 | 69.79 | 68.29 | 69.53 | 54.74 |
| Cambrian-1* | 8B | 55.60 | 32.53 | 35.39 | 43.46 | 49.14 | 78.15 | 67.22 | 51.64 |
| SPHINX* | 7B | 53.13 | 33.93 | 52.27 | 50.96 | 71.23 | 85.07 | 73.10 | 59.96 |
| LLaVA-NEXT-Interleave | 7B | 57.64 | 33.79 | 47.72 | 51.84 | 67.93 | 81.39 | 74.91 | 59.32 |
| InternLM-XComposer2-VL | 7B | 55.85 | 41.80 | 48.27 | 57.52 | 76.60 | 74.34 | 77.75 | 61.73 |
| LLaVA-OneVision | 7B | 51.77 | 46.13 | 41.85 | 62.19 | 69.73 | 90.31 | 80.93 | 63.27 |
| MiniCPM-V2.6 | 8B | 57.31 | 49.22 | 43.28 | 65.86 | 75.24 | **92.02** | 80.80 | 66.25 |
| InternVL2 | 8B | 59.97 | 43.85 | 47.91 | 57.60 | 78.10 | 74.18 | 80.37 | 63.14 |
| LLaVA-1.5 | 13B | 49.96 | 38.78 | 46.17 | 58.17 | 73.09 | 73.62 | 70.98 | 58.68 |
| LLaVA-NeXT | 34B | 57.92 | 48.79 | 52.87 | 71.34 | 80.28 | 81.12 | 77.80 | 67.16 |
| InternVL2 | 76B | **68.25** | **54.22** | **56.66** | **66.30** | **80.47** | 86.40 | **82.92** | **70.75** |

**Expert Agent.** Tool agent is an independent module within a large model, responsible for specific functions (Xi et al., 2023). In the evaluation of MMAD, exploring whether expert models can enhance MLLMs is an intriguing topic. We designed a simple method to investigate its feasibility. As shown in Figure 4(b), we treat the IAD model as a visual expert model. Since the output anomaly map is difficult to understand by MLLMs, we visualize it and then input it into the MLLMs. We use SOTA models in IAD, such as AnomalyCLIP and PatchCore, as experts and test the differences among 3 visualization methods. Additionally, we applied a special expert, ground truth, which directly uses the mask as the output of the expert model to verify the upper limit.

## 4 EXPERIMENTS

### 4.1 EVALUATION SETTINGS

We default to a 1-shot setting, where, in addition to the query image, we randomly provide a normal image from the dataset and inform the model that it can use this image as a template. The template image helps the model understand the normal state of objects, which is essential for anomaly detection. We also provide 0-shot and few-shot settings for further comparison. Besides, randomly providing normal images may lead to misunderstandings in some object categories due to multiple normal states existing. We introduce a 1-shot$^+$ setting, which retrieves the most similar image from the normal data as a template. This setting tests the model's ability to perform pairwise comparisons of industrial images.

For commercial models, we conducted tests using APIs, specifically versions claude-3-5-sonnet-20241022, gemini-1.5-flash-001, gemini-1.5-pro-001, gpt-4o-mini-2024-07-18, and gpt-4o-2024-05-13. For open-source models, we adapted and tested AnomalyGPT (Gu et al., 2024), Qwen-VL-Chat (Bai et al., 2023), LLaVA-1.5 (Liu et al., 2023a), Cambrian-1 (Tong et al., 2024a), SPHINX (Lin et al., 2023), LLaVA-NeXT (Liu et al., 2024a), LLaVA-NEXT-Interleave Li et al. (2024b), InternLM-XComposer2-VL (Dong et al., 2024), MiniCPM-V2.6 (Yao et al., 2024), and InternVL2 (Chen et al., 2023). We endeavored to maintain consistency with the default hyper-parameters and prompt methods provided. For models exceeding 20B parameters, we employed appropriate quantization methods. Each image's multiple VQA tasks were tested independently, and caches were cleared after each test. Some models only support single-image input by default. For LLaVA-1.5, we modified the framework to support multi-image input, while for Cambrian-1 and SPHINX, we only tested single-image input (i.e., 0-shot) performance. We will randomize the letters and order of the options and use the accuracy of responses as a metric. If the model does not

Table 3: Performance comparison with and without template images. (1-shot$^+$ indicates using a retrieved image as the template to provide more helpful visual information.)

| Model | Scale | Setting | Anomaly | Defect | | | | | Average |
| --- | --- | --- | --- | --- | --- | --- | --- | --- | --- |
| | | | Discrimination | Classification | Localization | Description | Analysis | | |
| Gemini-1.5-flash | - | 0-shot | 58.43 | 49.93 | 53.11 | 63.07 | 82.83 | | 68.58 |
| | | 1-shot$^+$ | 62.22 (+3.79) | 52.29 (+2.37) | 50.99 (-2.12) | 65.52 (+2.45) | 83.41 (+0.59) | | 69.59 (+1.01) |
| LLaVA-1.5 | 7B | 0-shot | 50.79 | 36.94 | 35.94 | 50.86 | 70.45 | | 54.91 |
| | | 1-shot$^+$ | 50.99 (+0.21) | 37.14 (+0.20) | 35.63 (-0.31) | 50.75 (-0.11) | 70.01 (-0.44) | | 54.66 (-0.25) |
| LLaVA-1.5 | 13B | 0-shot | 49.98 | 38.77 | 48.64 | 57.65 | 72.47 | | 58.58 |
| | | 1-shot$^+$ | 49.97 (-0.01) | 39.44 (+0.67) | 45.22 (-3.42) | 58.64 (+0.99) | 73.28 (+0.81) | | 58.55 (-0.02) |
| LLaVA-NeXT | 34B | 0-shot | 60.25 | 51.57 | 55.49 | 71.62 | 80.43 | | 68.45 |
| | | 1-shot$^+$ | 56.80 (-3.45) | 48.87 (-2.70) | 52.80 (-2.69) | 70.82 (-0.79) | 80.00 (-0.43) | | 67.16 (-1.28) |
| LLaVA-NEXT-Interleave | 7B | 0-shot | 58.39 | 36.98 | 48.98 | 51.51 | 66.64 | | 60.04 |
| | | 1-shot$^+$ | 57.71 (-0.68) | 35.87 (-1.11) | 48.11 (-0.86) | 52.40 (+0.89) | 67.88 (+1.24) | | 59.82 (-0.22) |
| InternLM-XComposer2-VL | 7B | 0-shot | 58.33 | 43.10 | 54.56 | 57.84 | 75.30 | | 62.78 |
| | | 1-shot$^+$ | 55.87 (-2.46) | 42.76 (-0.34) | 49.68 (-4.88) | 58.03 (+0.20) | 76.52 (+1.23) | | 61.99 (-0.79) |
| InternVL2 | 40B | 0-shot | 66.37 | 48.54 | 53.39 | 64.05 | 79.01 | | 69.23 |
| | | 1-shot$^+$ | 67.69 (+1.32) | 51.15 (+2.61) | 55.81 (+2.42) | 66.81 (+2.76) | 79.83 (+0.82) | | 70.71 (+1.49) |
| InternVL2 | 76B | 0-shot | 64.30 | 51.19 | 54.20 | 63.46 | 79.92 | | 69.41 |
| | | 1-shot$^+$ | 69.23 (+4.93) | 54.69 (+3.50) | 57.20 (+3.00) | 66.75 (+3.29) | 80.46 (+0.54) | | 71.31 (+1.90) |

provide any option, we will automatically match the closest option to the output as the answer. It is worth noting that, in the anomaly discrimination subtask, due to the imbalance distribution, we will separately calculate the accuracy of normal and abnormal samples and then use their mean as the final accuracy.

## 4.2 EXPERIMENTAL RESULTS

We compare the performance of over a dozen models, including commercial APIs, interleaved MLLMs, industrial MLLMs, and vision-centric MLLMs, as shown in Table 2. All models outperform the random baseline. The open-source models perform the best, with the average accuracy of the GPT-4o and Gemini-1.5-pro models reaching 74.9% and 73%, respectively. However, their cost-efficient counterparts, GPT-4o-mini and Gemini-1.5-flash, only achieved 66.3% and 68.9%, respectively, falling short of the best open-source model, InternVL2-76B, which achieved 70.8%. AnomalyGPT performs poorly overall, primarily due to its training on the IAD task in a fixed question-and-answer format, leading to severe overfitting issues. It demonstrates decent performance in anomaly discrimination because we specifically adapted the question format to suit its training. Similarly, the vision-centric MLLMs, Cambrian-1 and SPHINX, do not exhibit superior performance on the fine-grained visual tasks of MMAD, likely due to their foundational language models not being advanced enough. Among the general open-source MLLMs, earlier models like Qwen-VL-Chat and LLaVA-1.5 underperform compared to newer models like LLaVA-OneVision and MiniCPM-V2.6, indicating that advancements in general capabilities benefit performance on IAD tasks. MMAD uses a default 1-shot format, providing a normal image for comparison with the test image. Thus, multi-image understanding, especially image comparison, is crucial, while LLaVA-NEXT-Interleave, trained for this, does not perform outstandingly. LLaVA-NeXT-34B and InternVL2-76B, due to their larger scales, achieve the top two performances among open-source models, highlighting the importance of model size.

**Human evaluation.** We conduct a preliminary human evaluation using 177 examples randomly sampled from the entire benchmark. Eight evaluators were divided into two groups: 3 industrial anomaly detection researchers as experts and 5 ordinary participants. As shown in Table 2, ordinary outperform the best models by  4%, and experts by  12%. Humans excel in anomaly and defect-related tasks, highlighting MMAD's challenges and MLLMs' limitations in anomaly detection. However, in object-related VQA, GPT-4o surpasses humans due to its broader knowledge base, with examples provided in the appendix B.

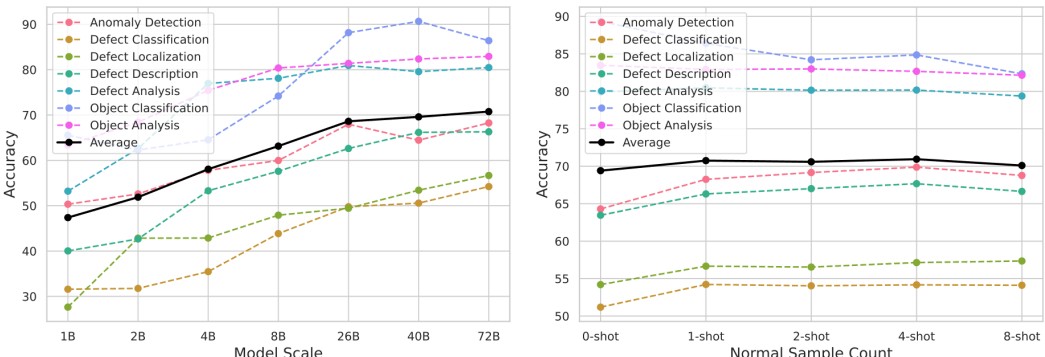

Figure 5: Left: Scaling law of model size in MMAD evaluation. We use the InternVL2 series as examples. Right: The performance trends of InternVL2-76B with varying counts of normal samples.

## 4.3 FURTHER ANALYSES

**Can MLLMs effectively utilize template normal images?** Anomaly detection often benefits from template images to understand normal patterns. To examine whether MLLMs can utilize templates, we conducted a comparative experiment. As shown in Table 3, we evaluated models in 0-shot and 1-shot+ settings. In 0-shot, no template image is provided, while in 1-shot+, the closest normal image to the test image is used as a template for comparison. Results indicate that most models fail to leverage template information, often resulting in performance drops. Due to cost constraints, we tested only the cost-effective Gemini-1.5-flash among commercial models, which showed notable gains, suggesting open-source models excel in contextual image understanding. Among open-source models, only the larger-scale InternVL2 effectively utilized template information.

**How significant is the impact of model scale on performance?** In the main experiment results, we observed that larger MLLMs generally exhibit better performance. We evaluate the scaling law using the InternVL2 series models, which utilize the same training data and are among the best-performing open-source models. As shown in the left panel of Figure 5, performance significantly improves with increasing model size, with the average accuracy difference between the largest and smallest models reaching 23.37%. Specifically, the classification performance of industrial objects improves the most with increasing size, and there are notable enhancements in several subtasks related to anomalies and defects. Although the performance improvement trend slows as the model size increases, further enlarging the model remains a promising option, albeit one that must be balanced against cost considerations.

**Can increasing the number of images further enhance performance?** For traditional few-shot IAD models, increasing the number of normal samples significantly enhances the performance of anomaly detection and localization. Therefore, we investigate whether MLLMs can leverage additional normal images. Experiments with InternVL2-76B (Figure 5, right) show performance improves from 0- to 1-shot, but further increases yield minimal gains (only slight improvements in anomaly/defect subtasks up to 4-shot). Beyond 8-shot, the performance in some subtasks even begins to decline, possibly due to information overload caused by too many input images, leading to confusion in the MLLMs. After providing multiple normal samples, the model needs to first summarize the normal characteristics and then compare them with the samples under inspection, while such tasks rarely appear in current MLLM instruction tuning data. This highlights the need for future research focusing on industrial applications' unique requirements.

## 4.4 EXPLORATION

**Input Domain Knowledge to MLLMs.** One significant challenge identified in our evaluation is the lack of industrial knowledge in MLLMs for specific tasks. During the training, MLLMs rarely encounter knowledge related to industrial quality inspection, while different products correspond to different specific knowledge. In practical applications, senior experts typically train workers. We use RAG to provide domain knowledge guidance to MLLMs to simulate this process. As shown in Table

Table 4: Performance comparison of different MLLMs with and without RAG.

| Model | RAG | Anomaly | Defect | | | | Object | | Average |
|---|---|---|---|---|---|---|---|---|---|
| | | Discrimination | Classification | Localization | Description | Analysis | Classification | Analysis | |
| LLaVA-NeXT-34B | - | 57.92 | 48.79 | 52.87 | 71.34 | 80.28 | 81.12 | 77.80 | 67.16 |
| | ✓ | 56.72 (-1.19) | 68.22 (+19.43) | 57.36 (+4.49) | 73.12 (+1.79) | 82.24 (+1.96) | 91.78 (+10.66) | 81.35 (+3.55) | 72.97 (+5.81) |
| InternVL2-26B | - | 67.96 | 49.77 | 49.44 | 62.62 | 80.92 | 88.16 | 81.39 | 68.61 |
| | ✓ | 68.64 (+0.68) | 67.32 (+17.55) | 53.81 (+4.37) | 70.84 (+8.22) | 82.18 (+1.26) | 93.81 (+5.64) | 83.31 (+1.92) | 74.27 (+5.66) |
| InternVL2-40B | - | 64.45 | 50.57 | 53.42 | 66.17 | 79.56 | 90.65 | 82.36 | 69.59 |
| | ✓ | 70.01 (+5.56) | 70.09 (+19.53) | 56.89 (+3.47) | 73.29 (+7.12) | 83.26 (+3.70) | 96.50 (+5.85) | 84.41 (+2.05) | 76.35 (+6.75) |
| InternVL2-76B | - | 68.25 | 54.22 | 56.66 | 66.30 | 80.47 | 86.40 | 82.92 | 70.75 |
| | ✓ | 66.68 (-1.57) | 70.95 (+16.73) | 60.57 (+3.91) | 75.32 (+9.02) | 82.71 (+2.24) | 91.71 (+5.31) | 85.29 (+2.36) | 76.18 (+5.43) |
| InternVL2-76B (0-shot) | - | 64.30 | 51.19 | 54.20 | 63.46 | 79.92 | 89.34 | 83.48 | 69.41 |
| | ✓ | 63.46 (-0.84) | 71.50 (+20.31) | 59.11 (+4.92) | 74.44 (+10.97) | 82.47 (+2.55) | 91.78 (+2.44) | 84.83 (+1.35) | 75.37 (+5.96) |

4, the inclusion of RAG significantly improves performance on MMAD across multiple models, with the most notable improvements in anomaly classification and object classification. This is primarily because domain knowledge contains extensive category information. The performance in defect localization also improves, indicating that models can leverage textual knowledge to enhance their perception of images. Notably, the performance in anomaly discrimination shows substantial improvement for InternVL2-40B, surpassing all other models in this metric.

**Model Collaboration.** Given that MLLMs have not been specifically trained on IAD data, their anomaly detection capabilities are relatively weak. Detecting anomalies is not particularly challenging for existing IAD models. Therefore, we are interested in whether MLLMs can perform answering with the help of the visual outputs of expert models. As shown in Table 6, we tested three agents: using AnomalyCLIP, Patch-Core, and directly using ground truth (GT) as experts, while also experimenting with various visualization methods. The two IAD models only

Figure 6: Performance comparison of different agent and visualization types. Based model is InternVL2-40B.

| Expert Model | Visualization Type | Anomaly | Defect | | |
|---|---|---|---|---|---|
| | | Discrimination | Classification | Localization | Description |
| baseline | - | 63.84 | 51.58 | 52.94 | 66.43 |
| AnomalyCLIP | bbox | -5.83 | -0.35 | +2.66 | -2.63 |
| AnomalyCLIP | contour | -6.40 | -1.19 | +2.52 | -3.86 |
| AnomalyCLIP | highlight | -4.21 | -6.92 | +7.44 | -6.61 |
| PatchCore | bbox | -9.58 | +0.10 | +1.00 | -0.51 |
| PatchCore | contour | -12.62 | -0.02 | -0.48 | -2.01 |
| PatchCore | highlight | -3.87 | -7.23 | +5.01 | -5.72 |
| GT | bbox | +10.10 | +7.10 | +21.29 | +5.40 |
| GT | contour | +10.86 | +6.45 | +20.44 | +6.16 |
| GT | highlight | +9.95 | +1.46 | +16.59 | +1.21 |
| GT | mask | -1.05 | +0.69 | +8.34 | +2.16 |
| GT | text | +24.89 | +1.36 | +28.01 | -0.22 |

improved defect localization but resulted in declines in performance for anomaly discrimination, defect classification, and defect description. In contrast, using GT improved all performance metrics, with the enhancement in defect localization far surpassing that of the two expert models. This indicates that the outputs of the current expert models are not sufficiently accurate to be used by MLLMs. Additionally, expert models will output anomaly maps for normal images, leading to overkill in anomaly discrimination. Among the various visualization methods, a simple bounding box (bbox) to highlight the anomaly region proved to be the most effective approach overall. Directly converting the GT mask into textual descriptions of defect location and size resulted in better performance in both discrimination and localization.

## 5 CONCLUSION

In this work, we introduce MMAD, the first benchmark for MLLMs in the IAD field, aimed at exploring the feasibility of using MLLMs for industrial quality inspection. Our evaluation of over ten SOTA MLLMs yielded less than optimistic results, revealing their weaknesses in industrial scenarios, particularly the lack of industrial knowledge and the ability to perform fine-grained comparisons across multiple images. Models may require extensive data for targeted improvement. Moreover, our further investigations suggest that MLLMs have the potential to address some of these issues through additional enhancements. We hope that MMAD will inspire future research into improving the relevant capabilities of MLLMs and promote the development of practical applications.

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

# A APPENDIX

## A.1 DISCUSSION OF SETTINGS

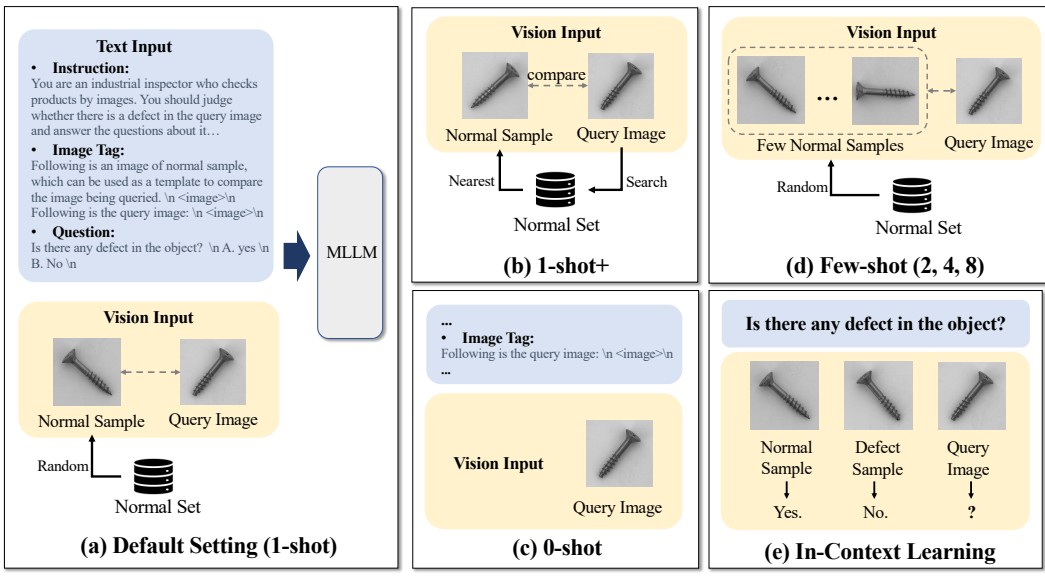

Figure 7: Diagram of different settings. The query image shows an abnormal screw with a manipulated front, which is difficult to identify without a compared sample.

Unlike other benchmarks, MMAD features various testing settings to simulate different scenarios in industrial quality inspection for MLLMs. We use 1-shot as the default setting, as a single normal image theoretically provides significant information compared to 0-shot. This approach is also used in our data annotation process. Increasing the number of normal images in few-shot settings may pose challenges for some MLLMs, particularly those not fine-tuned for multi-image tasks. In industrial production lines, the number of available normal samples often exceeds the context length of MLLMs. In such cases, similarity retrieval can filter out irrelevant images, leading us to propose the 1-shot$^+$ setting, where the most informative image is selected as a template. Additionally, abnormal images can sometimes be collected in production lines. To determine whether these abnormal images aid MLLMs in making judgments, we introduce abnormal image prompts in the 1-shot setting and then query the test image, termed in-context learning, as illustrated in Figure 7.

## A.2 ANALYSIS WITH IN-CONTEXT LEARNING

As shown in Table 5, we tested the performance under the in-context learning setting and compared it with other settings. We chose InternVL2-40B as the base model due to its superior context-handling abilities demonstrated in previous tests. The experimental results indicate that adding abnormal images in the in-context learning setting indeed helps MLLMs in anomaly detection, improving the performance from 63.84% to 66.47%. However, adding one more normal image does not significantly change this performance. In subsequent sub-tasks related to defect identification, in-context learning did not seem to provide any benefit. We believe this may be due to MLLMs not fully leveraging the information from abnormal images. Marking defects in abnormal images and combining them with textual prompts might better assist MLLMs. The 1-shot$^+$ setting, on the other hand, showed comprehensive improvements across sub-tasks, indicating that the quality of prompts is far more important than their quantity.

## A.3 ANALYSIS OF CHAIN OF THOUGHT

Chain of Thought (CoT) is a commonly used method to enhance the logical reasoning abilities of MLLMs (Chu et al., 2024). To investigate whether current MLLMs lack reasoning when addressing MMAD problems, we introduced a straightforward CoT approach. The process involves three

Table 5: Performance comparison of different settings in MMAD based on InternVL2-40B.

| Setting | Anomaly | Defect | | | | Object | | Average |
|---|---|---|---|---|---|---|---|---|
| | Discrimination | Classification | Localization | Description | Analysis | Classification | Analysis | |
| 1-shot | 63.84 | 51.58 | 52.94 | 66.43 | 79.75 | 90.76 | 82.41 | 69.67 |
| 1-shot+ | **67.81** | 54.18 | 55.99 | 68.14 | 81.75 | 90.76 | 82.98 | 71.66 |
| 2-shot | 63.75 | 50.38 | 54.36 | 67.22 | 79.54 | 90.49 | 81.85 | 69.66 |
| In-Context Learning | 66.47 | 50.56 | 51.78 | 64.87 | 78.55 | 91.33 | 81.65 | 69.32 |

Table 6: Performance comparison of different MLLMs with and without CoT.

| Model | CoT | Anomaly | Defect | | | | Object | | Average |
|---|---|---|---|---|---|---|---|---|---|
| | | Discrimination | Classification | Localization | Description | Analysis | Classification | Analysis | |
| InternVL2-40B | - | 64.45 | 50.57 | 53.42 | 66.17 | 79.56 | 90.65 | 82.36 | 69.59 |
| | ✓ | 59.42 | 46.10 | 51.92 | 57.66 | 79.80 | 80.55 | 85.29 | 65.82 |
| InternVL2-76B | - | 68.25 | 54.22 | 56.66 | 66.30 | 80.47 | 86.40 | 82.92 | 70.75 |
| | ✓ | 68.18 | 54.61 | 58.64 | 68.89 | 79.95 | 90.51 | 85.25 | 72.29 |

steps: first, the model identifies objects in the image; second, it compares the differences between the template image and the query image; and finally, it determines whether the identified difference constitutes a defect in the object. We incorporated a set of rules and adjusted the instructions accordingly, dividing the CoT responses into two stages. As shown in the Table 6, InternVL2-76B, the best-performing open-source MLLM we tested, achieved a 1.5% improvement in CoT-based performance. However, its anomaly detection accuracy, which is the most critical metric, showed no improvement. On the other hand, InternVL2-40B experienced a performance decline after introducing CoT, potentially due to insufficient stability in the language model's reasoning capabilities.

## A.4 ANALYSIS OF VISION DISABLE

Some studies (Tong et al., 2024a; Chen et al., 2024a) have proposed that determining whether a benchmark requires visual input to be solved has been a persistent challenge in vision-language research. To validate MMAD, we masked the visual components and compared the performance of MLLMs with and without visual input. The results, as shown in the Table 7, indicate that most subtasks in MMAD are highly dependent on the visual components. Performance significantly decreases when the visual input is removed.

## A.5 DETAILS OF HUMAN SUPERVISION

Our textual data is generated by MLLM, so its accuracy requires human supervision. In our designed process, human supervision is responsible for filtering the final model-generated multiple-choice questions. We first perform preliminary filtering through the program and then enable annotators to conduct an item-by-item review. A total of 26 personnel were involved in the review process, all of whom are researchers in industrial inspection or computer vision and possess a certain level of expertise. However, it should be noted that the industrial inspection field is highly specialized, with significant differences between products. Our annotators' understanding may differ from the professionals' understanding of where the original image data originates. We have developed a tool to enable annotators to filter out problematic multiple-choice questions quickly. As illustrated in Figure 8, we provide the original annotation information through visual and textual means to help annotators accurately identify objects and defects. At the same time, annotators do not need to correct every issue; they simply mark the erroneous questions for exclusion, significantly improving efficiency.

## A.6 DIVERSITY ANALYSIS OF DATASET

To systematically analyze the diversity of the dataset, we separately calculated the frequency of phrases appearing in questions and options, presenting them in the form of word clouds in Figure 9. In the question text, "defect" appeared most frequently, followed by "object", reflecting that our benchmark is constructed for industrial inspection scenarios, focusing on seven sub-tasks. In addition to common words such as "image", "type", and "appearance", the questions also include

Table 7: Performance comparison of different MLLMs with and without vision.

| Model | Vision | Anomaly | Defect | | | | Object | | Average |
|---|---|---|---|---|---|---|---|---|---|
| | | Discrimination | Classification | Localization | Description | Analysis | Classification | Analysis | |
| InternVL2-8B | Enable | 59.97 | 43.85 | 47.91 | 57.60 | 78.10 | 74.18 | 80.37 | 63.14 |
| | Disable | 49.71 | 32.28 | 41.38 | 52.20 | 75.34 | 41.94 | 57.13 | 50.00 |
| InternVL2-40B | Enable | 63.84 | 51.58 | 52.94 | 66.43 | 79.75 | 90.76 | 82.41 | 69.67 |
| | Disable | 49.61 | 24.43 | 36.35 | 46.39 | 68.05 | 37.29 | 55.75 | 45.41 |
| InternVL2-76B | Enable | 68.25 | 54.22 | 56.66 | 66.30 | 80.47 | 86.40 | 82.92 | 70.75 |
| | Disable | 49.93 | 25.92 | 28.54 | 43.88 | 70.41 | 35.89 | 57.53 | 44.59 |

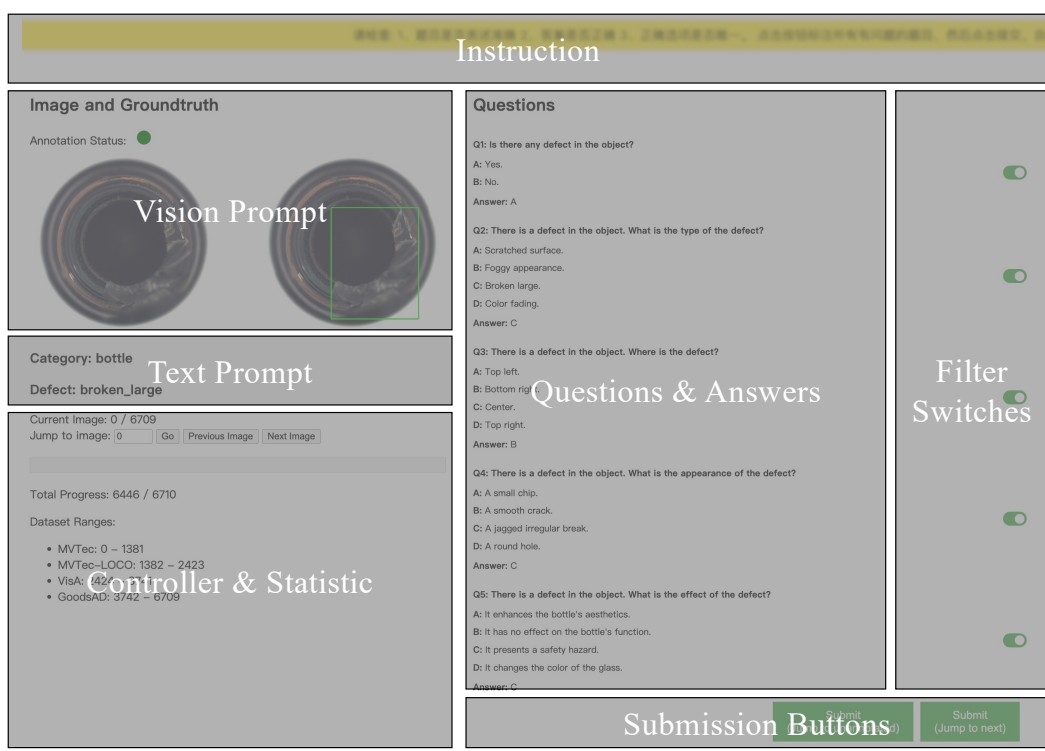

Figure 8: Illustration of human filtering tool. The tool comprises several functional areas. We use green bounding boxes to highlight defects as vision prompts, and we provide rough text prompts for object categories and defect types.

diverse expressions with lower frequencies. For example, expressions indicating position such as "relative position", "arrangement", and "defect located". In the text of the options, "Yes" and "No" are the standard answers for anomaly discrimination questions, thus appearing most frequently.

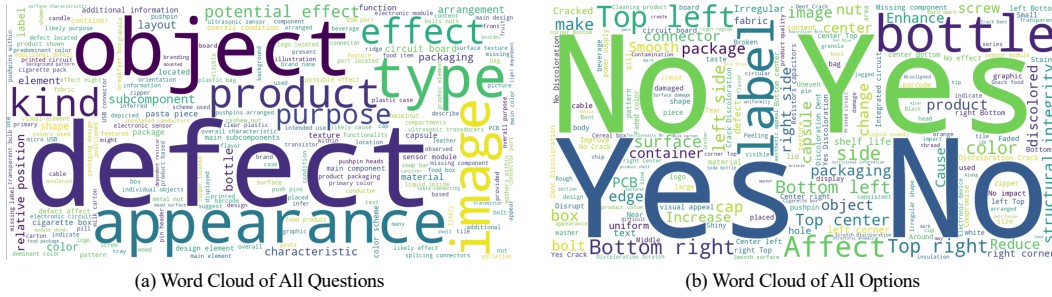

(a) Word Cloud of All Questions  (b) Word Cloud of All Options

Figure 9: Word frequency statistics presented in the form of a word cloud, with separate statistics for questions and options.

Beyond these two words, the diversity of options is more evident, including specific descriptions of various positions or expressions of different types of defects.

Table 8: Performance comparison of different MLLMs in Anomaly Discrimination/Detection Tasks.

| Model | Scale | Accuracy | Recall | Precision | F1 |
|---|---|---|---|---|---|
| Human (expert) | - | 95.24 | 94.25 | 98.89 | 96.43 |
| Human (ordinary) | - | 86.90 | 87.07 | 94.35 | 89.30 |
| claude-3.5-sonnet | - | 60.14 | 30.87 | 76.75 | 41.92 |
| Gemini-1.5-flash | - | 58.58 | 78.63 | 67.41 | 72.40 |
| Gemini-1.5-pro | - | 68.63 | 45.47 | 86.84 | 57.60 |
| GPT-4o-mini | - | 64.33 | 65.47 | 73.04 | 68.67 |
| GPT-4o | - | 68.63 | 67.37 | 75.68 | 71.04 |
| AnomalyGPT | 7B | 65.57 | 82.11 | 74.45 | 76.68 |
| Qwen-VL-Chat | 7B | 53.65 | 43.95 | 65.39 | 47.28 |
| LLaVA-1.5 | 7B | 51.33 | 94.79 | 62.72 | 75.32 |
| Cambrian-1* | 8B | 55.60 | 22.28 | 74.10 | 31.85 |
| SPHINX* | 7B | 53.13 | 6.42 | 99.74 | 10.61 |
| LLaVA-NEXT-Interleave | 7B | 57.64 | 16.58 | 90.83 | 25.64 |
| InternLM-XComposer2-VL | 7B | 55.85 | 17.94 | 75.87 | 27.16 |
| LLaVA-OnVision | 7B | 51.77 | 4.90 | 78.19 | 9.10 |
| MiniCPM-V2.6 | 8B | 57.31 | 34.38 | 70.98 | 45.31 |
| InternVL2 | 8B | 59.97 | 30.25 | 79.22 | 41.23 |
| LLaVA-1.5 | 13B | 49.96 | 99.79 | 62.00 | 76.28 |
| LLaVA-NeXT | 34B | 57.92 | 46.27 | 69.98 | 54.44 |
| InternVL2 | 76B | 68.25 | 55.81 | 83.52 | 64.40 |

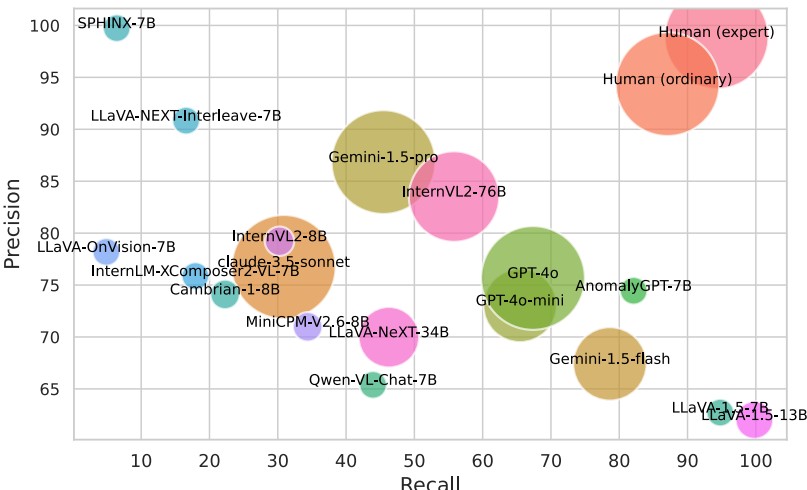

Figure 10: Bubble chart with recall and precision as axes in anomaly discrimination/detection task. The size of the bubbles represents the number of parameters in the model. For commercial models or humans where the number of parameters is unknown, the bubbles are set to the maximum size.

## A.7  COMPREHENSIVE METRICS FOR ANOMALY DISCRIMINATION/DETECTION TASKS

To comprehensively evaluate the performance of various models in the anomaly discrimination/detection task, we follow the traditional anomaly detection setup, treating the anomaly class as the positive class and the normal class as the negative class. We measured recall, precision, and F1-score. As shown in Table 8, by analyzing the recall and precision metrics, we can identify some reasons for poor accuracy performance in certain models. For instance, both SPHINX and LLaVA-OnVision have recall rates below 10%, indicating that these models frequently misclassify anomaly samples as normal, leading to a high rate of missed detections. On the other hand, LLaVA-1.5 has a high recall but low precision, suggesting a high rate of false positives. Humans, however, outperform MLLMs across all metrics, with human experts achieving over 94% and ordinary individuals

achieving over 87%. This disparity is visualized in the bubble chart, as shown in Figure 10, where there is a significant gap between the bubbles representing humans and those representing various MLLMs, and considerable differences among the MLLMs themselves. Additionally, it can be observed that the model AnomalyGPT, which is specifically trained for anomaly detection, performs better than most models but still suffers from a significant false positive issue.

## B  CASE STUDY

---

***Hit:***

*You are an industrial inspector who checks products by images. You should judge whether there is a defect in the query image and answer the questions about it.*
*Answer with the option's letter from the given choices directly!*
*The first image is the normal sample, which can be used as a template to compare.*
*The last image is the query image.*

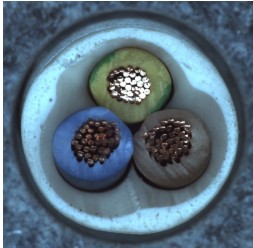 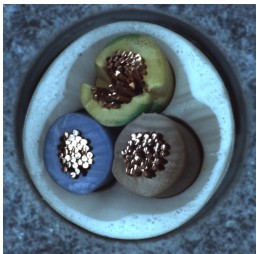

---

***Question:***

*Should the query picture be classified as normal or abnormal?*
*A.   Abnormal*
*B.   Normal*
*Give a short analysis and then answer a letter.*   ← only included in qualitative analysis

---

***InternVL-40B:***

*The query picture shows three objects with copper-colored spheres on top, which are not typical for a normal product. They appear to be some kind of artificial or experimental items rather than a standard product. Therefore, the query picture should be classified as abnormal.*
*Answer: A. Abnormal.*   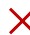

---

***GPT-4o:***

*Analysis: Comparing the query image to the normal template, the green section in the query image shows a visible crack or separation in its structure, which is not present in the normal image. This defect indicates that the product does not meet the standard criteria.*
*Answer: A*   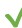

---

Figure 11: A case of a question-and-answer result in qualitative analysis. Since MMAD is testing with multiple-choice questions, we cannot directly analyze the answers. Instead, we use the chain-of-thought approach to encourage the model to provide its own analysis. In this example, the InternVL-40B model did not notice the damage to the cable, yet it still chose the correct answer from the options. In comparison, GPT-4o can output a proper analysis and letter. Subsequent issues such as anomaly classification, localization, and description can further test whether the model's correct answers are coincidental. Therefore, a comprehensive evaluation, rather than simple anomaly detection, is essential.

*Hit:*
*You are an industrial inspector who checks products by images. You should judge whether there is a defect in the query image and answer the questions about it.*
*Answer with the option's letter from the given choices directly!*
*The first image is the normal sample, which can be used as a template to compare.*
*The last image is the query image.*

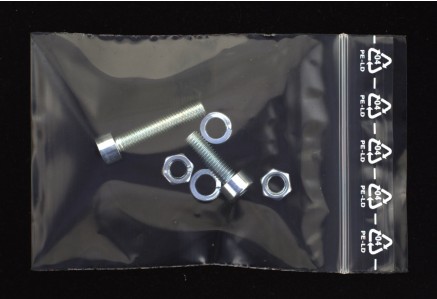 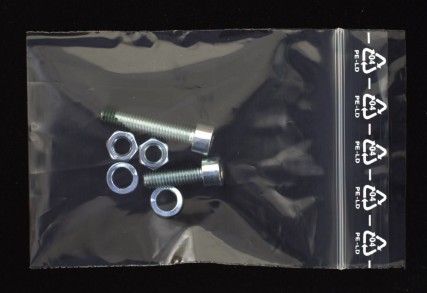

*Question:*
There is a defect in the object. What is the type of the defect?
 A: Incorrect labeling
 B: Structural deformation
 C: Missing component
 D: Discoloration
*Give a short analysis and then answer a letter.* ← only included in qualitative analysis

**GPT-4o:**
*Analysis: Comparing the query image to the normal sample, the query image is missing one washer. The normal sample has two bolts, two nuts, and two washers, whereas the query image has only one washer.*
*Answer: C: Missing component*  ✗

*Human:*
*Analysis:*
*It is easy to see that the end of the long screw in the inspected image has been stained green.*
*Answer: D: Discoloration*  ✓

Figure 12: A case of a question-and-answer result in qualitative analysis. A human can quickly notice the defect in the query image, while the model focuses on the number of components. This illusion tends to occur when the defect is minor, and the object's composition is complex.

*Hit:*

*You are an industrial inspector who checks products by images. You should judge whether there is a defect in the query image and answer the questions about it.*
*Answer with the option's letter from the given choices directly!*

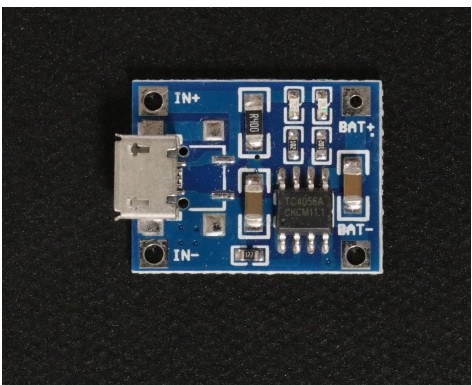

*Question:*
What is the purpose or function of this electronic circuit board?
A: Audio processing unit
B: Power supply for electronic devices
C: Lighting control system
D: Wireless communication module
*Give a short analysis and then answer a letter.* ← only included in qualitative analysis

*GPT-4o:*

*Analysis: This is a TP4056 module, commonly used as a lithium battery charger circuit. It features a micro-USB input (IN+) and battery connection points (BAT+ and BAT-). The presence of components like the TC4056A IC and the layout indicates its use in charging and regulating power supply for lithium-ion batteries.*
*Answer: B* ✓

*Human:*
*I don't know. It looks like all the options are possible.* ✗

Figure 13: A case of a question-and-answer result in qualitative analysis. For object-related questions, GPT-4o possesses extensive knowledge and can analyze object information based on subtle clues. However, most humans do not have such comprehensive knowledge. For instance, non-specialists may not recognize BAT+ and BAT-, let alone the TC4056A IC chip.

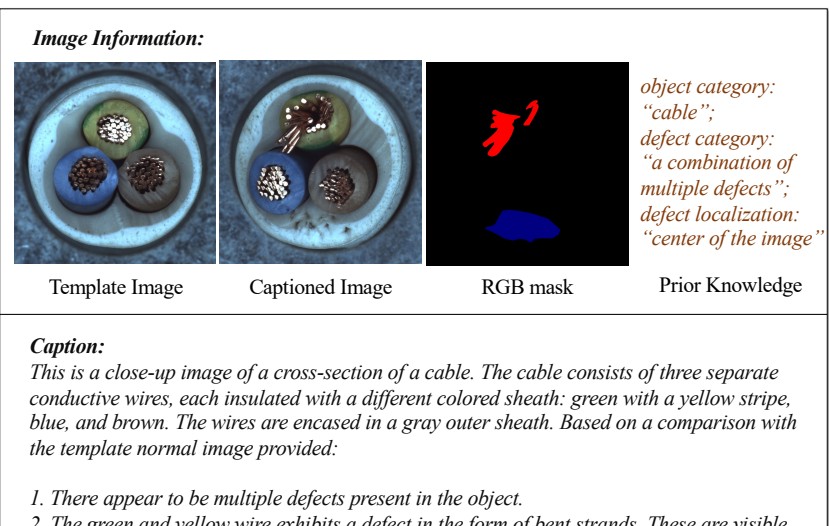

**Image Information:**

Template Image | Captioned Image | RGB mask | Prior Knowledge

*object category: "cable"; defect category: "a combination of multiple defects"; defect localization: "center of the image"*

**Caption:**
*This is a close-up image of a cross-section of a cable. The cable consists of three separate conductive wires, each insulated with a different colored sheath: green with a yellow stripe, blue, and brown. The wires are encased in a gray outer sheath. Based on a comparison with the template normal image provided:*

*1. There appear to be multiple defects present in the object.*
*2. The green and yellow wire exhibits a defect in the form of bent strands. These are visible at the approximate top center of the image, where the copper strands stick out at an irregular angle compared to the undisturbed arrangement seen in the normal image.*
*3. Another anomaly is present in the brown wire, where there seems to be damage to the insulation. This defect is located at the bottom center of the image, appearing as a deviation from the uniformly round shape of the corresponding wire in the normal sample.*

*These anomalies could potentially affect the performance and safety of the cable.*

Figure 14: A good case of image caption. We prompt the AI assistant that the image contains multiple defects and provide corresponding annotations. The AI assistant distinguishes different defects in the caption, describing their appearance and location individually.

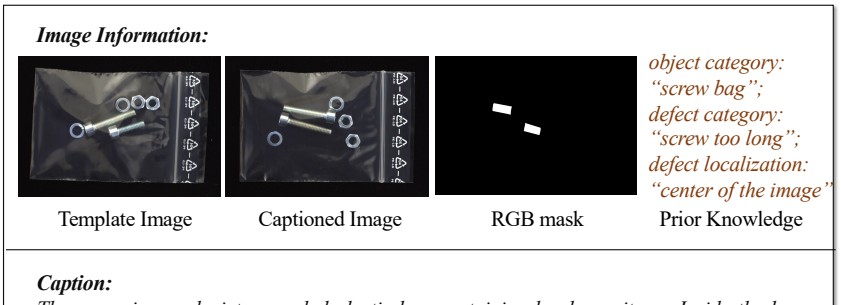

**Image Information:**

Template Image | Captioned Image | RGB mask | Prior Knowledge

*object category: "screw bag"; defect category: "screw too long"; defect localization: "center of the image"*

**Caption:**
*The query image depicts a sealed plastic bag containing hardware items. Inside the bag, there are two bolts, four nuts, and two washers. The bolts appear metallic with threads for fastening, and the nuts and washers accompany them as components for securing or distributing the load on the material being fastened.*

*Upon comparing the query image with the template normal image, there is a noticeable anomaly related to the length of the bolts. In the query image, one bolt, positioned in the bottom part of the bag, appears longer than the corresponding bolt in the normal image. Specifically, the defect is observed in the central bolt, which extends from the bottom left to the bottom center region of the bag. This anomaly disrupts the uniformity expected from a standard hardware set, where bolts should be of matching lengths for consistent application in construction or assembly tasks. The appearance of the anomaly is characterized by a visibly extended thread area beyond what is observed in the normal comparison image.*

Figure 15: A bad case of image caption. Although we informed the AI assistant that a screw is too long, it failed to analyze that one screw should be long and the other short. This error is primarily due to the insufficient number of template images, and such misunderstandings will be corrected during manual filtering.

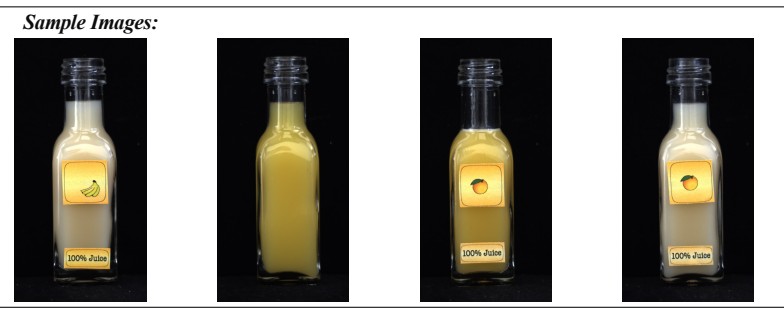

**Sample Images:**

**Domain Knowledge:**
*<Normal Characteristics>*
*Description: The ultrasonic sensor module, typically characterized by its rectangular blue printed circuit board (PCB), features two round ultrasonic transducers positioned symmetrically, which function as the transmitter and receiver. The module includes a central electronic component likely responsible for signal processing, surrounded by clear white labeling indicating each pin's function, such as VCC (power supply), Trig (trigger), Echo (echo signal), and GND (ground). Upon inspection, there are no visible defects or anomalies present; all components, including solder points, appear well-formed and intact. The noise features, which may include slight variations in texture or minor color differences, are typical in manufactured electronic components and do not indicate defects. The overall condition conveys a standard representation of a fully functional ultrasonic sensor in good working order, free from any damage, discoloration, or misalignment.*

Figure 16: A case of domain knowledge. Domain knowledge includes descriptions of the characteristics of normal samples. In this example, the text describes the components and appearance of this category of PCBs, as well as some common types of noise that may be present.

**Sample Images:**

**Domain Knowledge:**
*<Logical Anomalies>*
*Description: The observed defects primarily exhibit inconsistencies related to product labeling, fill levels, or the representation of contents within the juice bottles. Common visual features for distinguishing these logical anomalies include:*
*- Label Misalignment: Many images showed labels where graphical elements (such as fruit icons) were misplaced or not centered, deviating from their expected positions. This often causes confusion regarding branding and product identity.*
*- Missing Labels: Several instances indicated the absence of labels essential for branding, ingredient information, or flavor identification. Missing labels create visual gaps on the product, which can mislead consumers about the product's contents.*
*- Fill Level Deviations: Anomalies relating to insufficient or excessive juice levels compared to standard expectations were found. These incidents presented notable visual differences where expected liquid content was overflown, underfilled, or completely absent.*
*- Label Content Mismatch: Some products contained labels that represented a different fruit than the actual juice color, indicating a logical misrepresentation of the product's identity. This created a strong disconnect between the label imagery and the liquid contents inside the bottle.*
*All these features are significant in determining the presence and type of logical anomalies across the evaluated juice bottle batch.*

Figure 17: A case of domain knowledge. Domain knowledge includes descriptions of defects. In this example, the text outlines four major logical anomalies that may occur in these juice bottles. The actual dataset contains some rare logical anomalies that are not described, such as the incorrect icon placement in the first image. We do not expect domain knowledge to cover all possible anomalies; its primary role is to provide MLLMs with an understanding of the types of anomalies.

