# OpenReview forum: "MMAD: A Comprehensive Benchmark for Multimodal Large Language Models in Industrial Anomaly Detection"
_ICLR.cc/2025/Conference — ICLR 2025 Poster_

### Official Review · Reviewer_dwRW · 2024-10-21

**Soundness:** 3
**Presentation:** 3
**Contribution:** 3
**Rating:** 6
**Confidence:** 3

**Summary:**

The authors introduce MMAD, the first benchmark for evaluating Multimodal Large Language Models (MLLMs) in industrial anomaly detection (IAD). MMAD contains seven key subtasks relevant to industrial inspection with a dataset of 8,366 industrial images and 39,672 questions.

Their experiments show that while commercial models like GPT-4o perform best, achieving 74.9% accuracy, they still fall short of industrial requirements. Their analysis reveals that current MLLMs still have significant room for improvement in answering questions related to industrial anomalies and defects. The authors propose two training-free strategies to boost MLLM performance.

**Strengths:**

1. The paper is easy to follow.
2. The benchmark contains 39,672 questions for 8,366 industrial images, which is helpful for thoroughly evaluating the quality of MLLMs in an IAD setting.
3. Multiple LLMs were evaluated, and the training-free performance enhancement strategies appear to be useful.

**Weaknesses:**

1. The contribution are rather weak. While the authors introduce a new benchmark, it primarily relies on images from several existing benchmarks as seed data, using LLMs to generate a synthetic benchmark. This approach does not introduce significant advancements in terms of new methodologies or techniques. As such, claiming the benchmark as "first-ever comprehensive" is kinda misleading.
2. Moreover, the two enhancement strategies proposed are fairly standard and appear to be instantiations of existing solutions applied to the IAD domain without incorporating substantial “domain-specific” optimizations. Given the prevalence of similar techniques in the broader fields, I am skeptical about whether this paper will hold strong interest for the ICLR audience.

**Questions:**

Since the benchmark is generated from existing benchmarks, why is it described as the “first-ever comprehensive” benchmark?

---

> ### Author Response · Authors · 2024-11-25
>
> We believe the reviewer may have misunderstood the contribution of this paper. To clarify, we would like to restate the key contributions of MMAD as a benchmark. First, this benchmark fills the gap in the application of MLLMs in the industrial domain and sets new challenges for their capabilities. Second, we introduce a novel pipeline for generating semantic annotations for visual anomaly detection data, addressing the issue that MLLMs cannot be directly evaluated on IAD datasets. Finally, we comprehensively evaluate the performance of representative MLLMs on MMAD, highlight the weaknesses of current MLLMs in fine-grained industrial knowledge and multi-image understanding, and provide two general enhancement schemes.
>
> Now, we will address the reviewer’s comments one by one:
>
>
> ### 1. Weak Contribution Due to Dependence on Existing Datasets
>
> We believe that constructing a new benchmark based on existing datasets is a reasonable approach. On the one hand, industrial anomaly detection datasets have been extensively validated by researchers, providing a certain level of authority and reliability. On the other hand, most benchmarks for MLLMs are built upon existing datasets, provided that the images meet the evaluation criteria. Examples of such benchmarks include:
>
> - **MathVista** [1]
> - **Seed-Bench** [2]
> - **CompBench** [3]
> - **MME** [4]
> - **MME-Video** [5]
> - **MMBench** [6]
>
> In addition, existing industrial anomaly detection datasets cannot be directly used for evaluating MLLMs. We have designed a new annotation pipeline and incurred the cost of generating semantic annotations for the dataset. Furthermore, we have invested a substantial amount of manpower in filtering the data.
>
> ### 2. "First-ever comprehensive" is kind of misleading
>
> We believe the term "first-ever comprehensive" is not misleading. First, to the best of our knowledge, there is no existing benchmark in the industrial anomaly detection domain specifically designed for MLLMs, so the term "first-ever" is justified. Second, the comprehensiveness of our dataset is reflected not only in the variety of image categories and defect types but also in the breadth of sub-tasks. Existing datasets typically focus only on anomaly detection and localization. In contrast, our benchmark introduces additional tasks such as defect classification, defect analysis, defect description, object classification, and object analysis, resulting in a total of seven key sub-tasks.
>
>
> ### 3. Two enhancement strategies proposed are fairly standard
>
> The two enhancement strategies were conceived after evaluating MMAD, and they represent two potential avenues for improvement. RAG aims to test whether knowledge-based texts can assist in anomaly detection, while the Agent strategy tests whether existing expert models can help improve the perceptual capabilities of MLLMs. Our intention was not to develop entirely novel methods but rather to conduct experiments by applying existing ideas in a reasonable manner.
>
> However, we have “domain-specific” optimizations while we apply the concepts of RAG and Agent, and their specific implementations are entirely new. Specifically, the database construction in RAG is different from existing general methods. We created a retrieval library by summarizing defect and normal samples using descriptions generated by both experts and models rather than by collecting data from the web. Moreover, anomaly detection models have never been integrated into the Agent framework before, and we are the first to introduce such models. As a result, we also explored novel visualization methods for this approach.
>
>
> ### 4. Interest for the ICLR audience
>
> We believe that ICLR has a strong sense of inclusivity and a practical focus, which makes it highly relevant to the industrial anomaly detection domain. For example, works like AnomalyCLIP[7] and MuSc[8] have been well-received. Additionally, many other evaluation-focused works in different domains, such as MathVista[1] and ViLMA, have been presented at ICLR. Therefore, we argue that the contribution to ICLR is not solely based on presenting novel methods but also on advancing important application domains and benchmarks.

---

> > ### Author Response · Authors · 2024-11-25
> > **References**
> >
> > [1] Lu, P., Bansal, H., Xia, T., Liu, J., Li, C., Hajishirzi, H., Cheng, H., Chang, K., Galley, M., & Gao, J. (2023). MathVista: Evaluating Mathematical Reasoning of Foundation Models in Visual Contexts. International Conference on Learning Representations, ICLR 2024.
> >
> > [2] Li, B., Ge, Y., Ge, Y., Wang, G., Wang, R., Zhang, R., & Shan, Y. (2024). SEED-Bench: Benchmarking Multimodal Large Language Models. 2024 IEEE/CVF Conference on Computer Vision and Pattern Recognition (CVPR), 13299-13308.
> >
> > [3] Kil, J., Mai, Z., Lee, J., Wang, Z., Cheng, K., Wang, L., Liu, Y., Chowdhury, A.S., & Chao, W. (2024). CompBench: A Comparative Reasoning Benchmark for Multimodal LLMs. NeurIPS 2024.
> >
> > [4] Fu, C., Chen, P., Shen, Y., Qin, Y., Zhang, M., Lin, X., Qiu, Z., Lin, W., Yang, J., Zheng, X., Li, K., Sun, X., & Ji, R. (2023). MME: A Comprehensive Evaluation Benchmark for Multimodal Large Language Models. ArXiv, abs/2306.13394.
> >
> > [5] Fu, C., Dai, Y., Luo, Y., Li, L., Ren, S., Zhang, R., Wang, Z., Zhou, C., Shen, Y., Zhang, M., Chen, P., Li, Y., Lin, S., Zhao, S., Li, K., Xu, T., Zheng, X., Chen, E., Ji, R., & Sun, X. (2024). Video-MME: The First-Ever Comprehensive Evaluation Benchmark of Multi-modal LLMs in Video Analysis. ArXiv, abs/2405.21075.
> >
> > [6] Liu, Y., Duan, H., Zhang, Y., Li, B., Zhang, S., Zhao, W., Yuan, Y., Wang, J., He, C., Liu, Z., Chen, K., & Lin, D. (2023). MMBench: Is Your Multi-modal Model an All-around Player? European Conference on Computer Vision.
> >
> > [7] Zhou, Q., Pang, G., Tian, Y., He, S., & Chen, J. (2023). AnomalyCLIP: Object-agnostic Prompt Learning for Zero-shot Anomaly Detection. ICLR 2024.
> >
> > [8] Li, X., Huang, Z., Xue, F., & Zhou, Y. (2024). MuSc: Zero-Shot Industrial Anomaly Classification and Segmentation with Mutual Scoring of the Unlabeled Images. ICLR 2024.
> >
> > [9] Kesen, I., Pedrotti, A., Dogan, M., Cafagna, M., Acikgoz, E.C., Parcalabescu, L., Calixto, I., Frank, A., Gatt, A., Erdem, A., & Erdem, E. (2023). ViLMA: A Zero-Shot Benchmark for Linguistic and Temporal Grounding in Video-Language Models. ICLR 2024.

---

> > > ### Comment · Reviewer_dwRW · 2024-11-27
> > >
> > > Thank you for your detailed response. I appreciate the effort to clarify the contributions of MMAD and to address the concerns raised. However, after reviewing your rebuttal and comments from other reviewers, I still hold reservations and remain unconvinced that this work aligns with the scope and expectations of ICLR papers.
> > >
> > > Thank you as well for mentioning these works. While constructing benchmarks from existing datasets is a common practice, it does not inherently establish novelty. The examples you cite, such as MathVista and Seed-Bench, evaluate LLMs on foundational capabilities like mathematical reasoning and visual understanding. These domains are central to the development of LLMs and align directly with their claimed general-purpose capabilities.
> > >
> > > In contrast, MMAD focuses on industrial anomaly detection, a domain where most existing LLMs neither claim expertise nor are optimized for performance. This weakens the analogy to benchmarks such as MathVista and Seed-Bench.

---

> > > > ### Author Response · Authors · 2024-11-29
> > > > **Second Round Responds to Reviewer Comments - 1**
> > > >
> > > > Thank you for further comments, which helped me better understand your remaining concerns.
> > > >
> > > > First, let's clarify the points where we have reached agreement:
> > > > - Building new benchmarks from existing datasets is reasonable
> > > > - MMAD is indeed the "first-ever comprehensive" benchmark for MLLMs in IAD
> > > >
> > > > It seems that you still have questions about MMAD's novelty and its contributions to the scientific community, which we will address separately:
> > > >
> > > > ---
> > > > ### 1. Novelty
> > > > MMAD is the first to evaluate MLLMs' capabilities in industrial vision understanding systematically. We designed the necessary data and evaluation dimensions for comprehensive MLLM testing in IAD, employed new data generation methods, designed unique instructions to drive diverse Q&A generation, and manually filtered to ensure accuracy. We comprehensively evaluated the resulting dataset and performed in-depth analysis, yielding many new instructive conclusions. These are the main contributions discussed in our paper, which we won't enumerate individually again here.
> > > >
> > > >  Other reviewers generally acknowledged these novelties, for example:
> > > > - Reviewer nkBV: "It covers a wide range of tasks related to industrial anomaly detection."
> > > > - Reviewer 69ie: "They also contribute to the diversity and novelty of benchmark tasks."
> > > > - Reviewer YdC1: "Importantly, it seems to be filling an important gap in real-world MLLM evaluations, which I think makes for a strong claim of significance."
> > > >
> > > > Since the main content of our paper is to propose a new benchmark rather than a new method, we believe that the novelty can also be demonstrated through the completeness of the benchmark. Each new experiment conducted on such a new benchmark differs entirely from previous research, and the experimental results will reveal new conclusions. There are a few concerns from reviewers regarding the completeness. Based on feedback from other reviewers, we have further enhanced the completeness of this work:
> > > > - Added human baseline to comprehensively illustrate the gap between MLLMs and humans in industrial tasks (see Response 1 to Reviewer YdC1)
> > > > - Added details on human supervision and diversity analysis of the dataset to enhance the benchmark's credibility and effectiveness (see Responses 1 and 2 to Reviewer nkBV)
> > > > - Added comprehensive analysis. Added four sets of experimental results and qualitative analysis to enhance evaluation comprehensiveness (see Response 3 to Reviewer nkBV and Responses 3 and 5 to Reviewer 69ie)

---

> > > > ### Author Response · Authors · 2024-11-29
> > > > **Second Round Responds to Reviewer Comments - 2**
> > > >
> > > > ### 2. Contributions to Industry Field
> > > > The AD and IAD fields are crucial areas of research, and extensive applications of multimodal technologies are expected in the future. These fields are key components in enhancing production efficiency through AI, serving as practical applications where multimodal technologies can demonstrate their value. We believe that MLLMs have the potential to become a new approach to solving IAD problems. Compared to traditional IAD methods, MLLMs not only support flexible visual prompts but also allow for language prompts to assist in model judgment (these points have been validated through our experiments). We have proposed the first evaluation benchmark, which is of significant importance to the industrial field.
> > > >
> > > > Additionally, we have observed that there is currently very limited research on MLLMs in the field of industrial imaging. In contrast, a similar field—medical imaging—has made significant advancements in this research direction and has already gained widespread recognition from the ML community, such as LLM-CXR [1] and LLaVA-Med [2]. By introducing MMAD, we hope to draw more attention from the community in this direction, thereby driving the development of industrial productivity.
> > > >
> > > > ---
> > > > ### 3. Contributions to the General ML Community
> > > >
> > > > While MMAD focuses on industrial anomaly detection, it also has a unique value for both IAD and general domains, as discussed in our two rounds with Reviewer 69ie regarding "unique challenges" and "value to the ML community."
> > > >
> > > > To summarize the value of general MLLM development:
> > > > - Industrial Scenarios are Simple but Require High Requirements. The high requirements of industrial tasks mean that MLLMs solving IAD problems must possess more excellent stability.
> > > > - Unlike general visual Q&A tasks, which typically focus on object-level features, industrial anomaly detection emphasizes low-level visual characteristics. These low-level visual features are crucial for understanding the real world.
> > > > - IAD examines MLLMs' fine-grained image comparison abilities, which previous benchmarks rarely focused on. This capability is fundamental for AGI and applicable across many domains.
> > > >
> > > > In conclusion, we strongly believe that our work offers unique and valuable perspectives that deserve to be shared with the scientific community. We hope that the aforementioned responses highlight the potential contributions our work can make and encourage you to reconsider our submission. We look forward to the possibility of contributing to the ongoing discourse and advancement within our field.
> > > >
> > > > [1] Lee, Suhyeon, et al. "LLM-CXR: Instruction-Finetuned LLM for CXR Image Understanding and Generation." The Twelfth International Conference on Learning Representations, ICLR 2024.
> > > >
> > > > [2] Li, Chunyuan, et al. "Llava-med: Training a large language-and-vision assistant for biomedicine in one day." NeurIPS 2023.

---

> > > > > ### Comment · Reviewer_dwRW · 2024-11-30
> > > > >
> > > > > Thank you for your efforts during the rebuttal phase.
> > > > >
> > > > > However, I still have some concerns regarding the phrase “first-ever comprehensive,” as it appears obscure in the context.
> > > > >
> > > > > I would appreciate it if the authors could include a detailed discussion in the paper on the broader impact of the benchmark for the general machine learning community.
> > > > >
> > > > > I will update my score accordingly.

---

> > > > > > ### Author Response · Authors · 2024-12-01
> > > > > >
> > > > > > Thank you for your feedback and acknowledgment. Your comments have helped us identify areas for improvement in our paper.
> > > > > >
> > > > > > In future versions, we will optimize our phrasing and add a separate discussion on the broader impact of the benchmark for the general machine-learning community.

---

### Official Review · Reviewer_YdC1 · 2024-11-03

**Soundness:** 4
**Presentation:** 3
**Contribution:** 3
**Rating:** 8
**Confidence:** 4

**Summary:**

The authors introduce MMAD—the first large-scale industrial anomaly detection dataset, specifically targeted at evaluating multimodal large language models. The dataset contains seven main subtasks: anomaly discrimination, defect classification, object classification, defect localization, object analysis, defect description, and defect analysis, which are all highly relevant to the IAD problem. The authors evaluate multiple out-of-the-box MLLMs on their dataset, as well as using RAG and expert model-based improvements. Finally, the authors present a very thorough evaluation of model performance by analyzing whether template images help in anomaly detection, how model scale contributes, and how the number of images affects the results.

**Strengths:**

The proposed dataset is original, and seems relatively high-quality. Importantly, it seems to be filling an important gap in real-world MLLM evaluations, which I think makes for a strong claim of significance.

The paper is generally clear and well-written, and the figures are very well-constructed.

The thoroughness of the evaluations is impressive; I especially appreciate Table 3 and the corresponding analysis of template image usefulness. Additionally, I am surprised that RAG is this helpful—and I'm glad the evaluations are detailed enough to observe this effect! I also appreciate the mention of specific model versions in evaluation, such as "gpt-4o-mini-2024-07-18".

**Weaknesses:**

The paper does not present any human baseline—while every example undergoes manual inspection for a theoretical 100% accuracy, it would be interesting to know what industrial experts achieve. Additionally, several frontier MLLMs are not evaluated, such as Claude Sonnet 3.6.

Some minor nitpicks/thoughts on style (feel free to disregard):

- Line 42: "represented by" is a somewhat strange term—maybe "such as" would be more accurate?
- Line 43: I would maybe rephrase this; "finish" is an unclear term
- Line 49: probably "mechanisms"
- Line 50: either the citations should not be parenthetical here, or it's unclear to me what the subject of the sentence is
- Line 79: I'm unsure what this sentence means; what is "unfair" about the comparison?
- Line 81: I guess you have already introduced it, so "committed" is a somewhat strange word to use here (e.g., why not just "We introduce")
- Line 93: would add a space before the parenthetical
- Line 346: this is probably not the usual definition of "autonomous agent"

**Questions:**

I am not an expert in industrial anomaly detection; as such, I do not have a good intuition for what exactly the mechanism by which this dataset contributes to e.g. manufacturing actually is; the authors seem much more well-versed in this than I am, so it would be interesting to hear more detailed thoughts. Besides this, however, I do not have many questions, as the paper is relatively clear and self-contained.

---

> ### Author Response · Authors · 2024-11-25
>
> Thank you for your recognition and suggestions. We have made several additions to the paper, and below are our responses to each of your comments:
>
> ### 1. Add human baseline
>
> First, we would like to clarify that humans do not achieve 100\% accuracy in the MMAD evaluation. This is because during the benchmark construction, human annotators had access to the original data annotations as references. However, during the human evaluation, evaluators do not have access to this data, which means they might miss subtle flaws or make mistakes for other reasons.
>
> We have added detailed information about the human evaluation in Section 4.2. The updated text is as follows:
>
> *Human evaluation. We conduct a preliminary human evaluation using 177 examples randomly sampled from the entire benchmark. Eight evaluators were divided into two groups: 3 industrial anomaly detection researchers as experts and 5 ordinary participants. As shown in Table 2, ordinary participants outperform the best models by 4\%, and experts by 12\%. Humans excel in anomaly and defect-related tasks, highlighting MMAD's challenges and MLLMs' limitations in anomaly detection. However, in object-related VQA, GPT-4o surpasses humans due to its broader knowledge base, with examples provided in Appendix B.*
>
> Some of the data from Table 2 are as follows:
> |            Model           | Scale |    Anomaly Discrimination | Defect Classification | Defect Localization | Defect Description | Defect Analysis | Object Classification | Object Analysis | Average |
> |:--------------------------:|:-----:|:--------------:|:--------------:|:------------:|:-----------:|:--------:|:--------------:|:--------:|:-------:|
> |       Human (expert)       |   -   |     95.24      |     75.00      |    92.31     |    94.20    |  83.33   |     86.11      |  80.37   |  86.65  |
> |      Human (ordinary)      |   -   |     87.96      |     66.25      |    85.58     |    81.52    |  71.25   |     90.97      |  69.31   |  78.98  |
> | claude-3-5-sonnet-20241022 |   -   |     60.14      |     60.14      |    48.81     |    67.13    |  79.11   |     85.19      |  79.83   |  68.36  |
> |      Gemini-1.5-flash      |   -   |     58.58      |     54.70      |    49.10     |    66.53    |  82.24   |     91.47      |  79.71   |  68.90  |
> |       Gemini-1.5-pro       |   -   |     68.63      |     60.12      |    58.56     |    70.38    |  82.46   |     89.20      |  82.25   |  73.09  |
> |         GPT-4o-mini        |   -   |     64.33      |     48.58      |    38.75     |    63.68    |  80.40   |     88.56      |  79.74   |  66.29  |
> |           GPT-4o           |   -   |     68.63      |     65.80      |    55.62     |    73.21    |  83.41   |     94.98      |  82.80   |  74.92  |
>
> ### 2. Evaluate Claude Sonnet 3.5
>
> We have tested the latest Claude-3-5-Sonnet-20241022, and the results are also included in Table 2. However, this commercial model performed poorly, with results similar to the much cheaper Gemini-1.5-flash.
>
> ### 3. Response to Minor Style Suggestions
>
> We appreciate your valuable suggestions and have made the corresponding revisions. Here, we would like to clarify two specific changes:
>
> I. The term "unfair" in Line 79 refers to previous works [1] that forced MLLMs to output anomaly detection results (such as a heatmap), rather than testing their linguistic outputs, which we believe is an unfair comparison.
>
> II. The term "autonomous agent" in Line 346 was indeed not used correctly, and we have changed it to "Tool agent."
>
>
> ### 4. The Mechanism of Contribution to Manufacturing
>
> This dataset is primarily used to measure the capabilities of MLLMs and promote their advancement in industrial applications. Several sub-tasks mentioned in the paper are of significant interest in the industrial domain. If an MLLM performs well on the current dataset, it could potentially be deployed directly on the production line as a quality control worker. In this case, humans could provide information to the model in various forms, such as a few images, brief textual descriptions, or complex documents. MLLMs can not only perform quality control tasks but also combine other capabilities, such as data summarization, table generation, and more, to bring additional possibilities to the manufacturing industry.
>
>
> [1] Zhang, J., Chen, X., Xue, Z., Wang, Y., Wang, C., & Liu, Y. (2023). Exploring Grounding Potential of VQA-oriented GPT-4V for Zero-shot Anomaly Detection. ArXiv, abs/2311.02612.

---

> > ### Comment · Reviewer_YdC1 · 2024-11-27
> >
> > Thank you! This addresses all of my questions; I have raised my score for the paper. Thanks for the detailed responses

---

> > > ### Author Response · Authors · 2024-11-29
> > > **Thank You Very Much for Your Feedback**
> > >
> > > Dear Reviewer YdC1,
> > >
> > > Thank you for your swift response and the revised evaluation of our paper. We sincerely appreciate your dedicated time and effort in reassessing our work. Your initial feedback was instrumental in enhancing our paper.
> > >
> > > Should any additional improvements or unresolved issues impact the review, we would be grateful for your suggestions. We are committed to resolving any remaining concerns.
> > >
> > > Thank you once again for your support.
> > >
> > > Best regards,
> > > The Authors

---

### Official Review · Reviewer_69ie · 2024-11-04

**Soundness:** 3
**Presentation:** 3
**Contribution:** 2
**Rating:** 6
**Confidence:** 3

**Summary:**

This paper introduces the first benchmark for industrial anomaly detection with multimodal LLMs. It defines 7 subtasks for industrial anomaly detection and creates a novel pipeline for generating semantic annotations of visual anomaly detection data. From the experiments, the best performing model GPT-4o reaches 74.9% accuracy, but it is still way below the industry needs and demonstrates the potential for future research.

**Strengths:**

Overall, this paper is clear, well-motivated and provides a new benchmark for industrial anomaly detection with multimodal LLMs. The tasks selected from the benchmark are comprehensive for identifying industrial anomaly, including defect detection, classification, localization, description, analysis, etc. They also contribute to the diversity and novelty of benchmark tasks. In addition, the paper highlights a few explorational techniques that could improve model performance, such as RAG on domain knowledge, few shots learning and model collaboration. Those explorations are helpful for industrial practitioners using the benchmark in production.

**Weaknesses:**

The paper could benefit from discussions with regards to the following points:
1. This paper fills a gap in the specific domain of industrial anomaly detection by providing a dedicated Multimodal LLM benchmark, and the benchmark has clear value in real-life industrial quality inspections. However, it is not clear how the industrial anomaly detection tasks are fundamentally different from other general visual Q&A tasks, and how this benchmark creates significant contribution to the general ML community beyond its specific application in industrial anomaly detection. There are many other venues that are great fit for industry-specific benchmarks, such as application-focused workshops and NeurIPS dataset and benchmark track.
2. The benchmark constructs the 7 key subtasks in the form of multiple choices. This format could be limiting or misguiding, as it is dependent on how the choices are created, the number of choices provided to the model, and the model could provide the correct answer without truly understanding the answer. For instance, in the defect classification, location, and analysis tasks, accurate open-text answers and locations are much harder to generate and evaluate than multiple choices, but those formats are directly useful in production settings.
3. In addition, the benchmark only reports on accuracy based on multiple choices, but fails to report recall, F1 score, etc. which would be important in the anomaly detection setting.

**Questions:**

1. It is interesting to see how RAG improves the performance with domain knowledge, as anomaly detection is a multimodal task, not a direct text retrieval task. How could prompting with domain knowledge affect the model performance, as context windows are now getting larger?
2. It is interesting to use an expert model to assist MLLMs - are there other ways of providing the expert information beyond the visualizations listed such as highlight, contour and BBox? It is intuitive to see how BBox can massively improve localization, but wonder if other direct text or other modal information can improve the performance.

---

> ### Author Response · Authors · 2024-11-25
> **Rebuttal to Reviewer Comments - 1**
>
> Thank you for recognizing the contribution of our dataset. We will address your concerns one by one.
>
> ### 1. Contribution to the General ML Community
>
> We believe that MMAD also makes an important contribution to the general ML community for three main reasons:
>
> I **The Need for AGI to Handle Simple Tasks**: General Artificial Intelligence (AGI) should possess the ability to handle basic tasks. AGI should not only solve complex logical problems but also take over repetitive and simple tasks, much like the work of industrial quality inspectors. As we mentioned in the paper:
>
> > "Nowadays, multimodal large language models, represented by GPT-4, can already do many human jobs, especially high-paying intellectual jobs like programmers, writers, and data analysts. In comparison, the work of quality inspectors is simple, typically not requiring a high level of education but relying heavily on work experience. Therefore, we are greatly interested in the question: How well are current MLLMs performing as industrial quality inspectors?"
>
> However, our testing data reveals that, even for simple tasks, current MLLMs still require significant improvements.
>
> II **Broad Relevance to Anomaly Detection**: The data we tested uses anomaly detection tasks, which have broad relevance to other fields. For example, anomaly detection in medical imaging, remote sensing, and other areas could benefit from similar approaches and methodologies.
>
> III **Revealing Gaps in Current MLLM Capabilities**: Our benchmark introduces a new, specific application for MLLMs, while also highlighting areas of capability gaps that are rarely addressed in previous benchmarks. For instance, fine-grained image comparison is crucial in our benchmark, yet existing benchmarks like CompBench, while offering image comparison tasks, rely on synthetic tasks that do not provide realistic or actionable references.
>
> ### 2. Multiple Choice Format
>
> We acknowledge that the multiple-choice format differs from real-world scenarios, but we believe it is currently the most appropriate format for constructing MMAD, for the following three reasons:
>
> I. **Multiple-choice questions exist in industrial settings**. For instance, when asking whether there is an anomaly, the only options are typically "Yes" or "No". The number of defect categories is also limited, allowing the model to choose from a set of options. Furthermore, the position of the defect and the affected category can be designed in a limited manner, such as dividing the area into a grid (e.g., a 3x3 grid) or categorizing the impact by function.
>
> II. **Accuracy in multiple-choice questions reflects model performance**. The multiple-choice format is also adopted by many well-known general benchmarks, such as MMBench[1] and MMMU[2]. We provide only four options for the model to select from, which could indeed lead to situations where the model provides the correct answer without fully understanding it. An example of this is provided in Appendix B, Figure 10. However, our benchmark asks multiple questions from different angles about the image. The model must form a correct understanding to answer subsequent questions. Therefore, our benchmark is capable of comprehensively evaluating the model's abilities.
>
> III. **Open-text answers in industrial scenarios are difficult to evaluate**. In industrial-related question-answering, the correctness of open-text answers is hard to measure. For example, the same defect might be referred to by multiple names, or there may be proprietary terms. Thus, matching words becomes unfeasible, and at present, no suitable text embeddings for industrial domains exist. Additionally, MLLMs generally lack industrial domain knowledge, making it difficult to evaluate open-text answers in this context.
>
> [1] Liu, Yuanzhan et al. "MMBench: Is Your Multi-modal Model an All-around Player?" ECCV 2023.
> [2] Yue, Xiang et al. "MMMU: A Massive Multi-Discipline Multimodal Understanding and Reasoning Benchmark for Expert AGI." CVPR 2024.

---

> ### Author Response · Authors · 2024-11-25
> **Rebuttal to Reviewer Comments - 2**
>
> ### 3. Report Recall
>
> Given that other sub-tasks have four options and recall cannot be calculated, we assume that your concern is about recall in the anomaly discrimination task with two options. In this task, we use the average accuracy of normal and abnormal samples as a metric, which provides a good measure of the model's performance. Based on your suggestion, we further measured recall for each model in the anomaly discrimination task, as shown in the table below.
>
> |          Model         | Scale | Accuracy | Recall |
> |:----------------------:|:-----:|:--------:|:------:|
> |     Human (expert)     |   -   |  95.24   | 94.25  |
> |    Human (ordinary)    |   -   |  87.96   | 87.07  |
> |    claude-3.5-sonnet   |   -   |  60.14   | 30.87  |
> |    Gemini-1.5-flash    |   -   |  58.58   | 78.63  |
> |     Gemini-1.5-pro     |   -   |  68.63   | 45.47  |
> |       GPT-4o-mini      |   -   |  64.33   | 65.47  |
> |         GPT-4o         |   -   |  68.63   | 67.37  |
> |       AnomalyGPT       |   7B  |  65.57   | 82.11  |
> |      Qwen-VL-Chat      |   7B  |  53.65   | 43.95  |
> |        LLaVA-1.5       |   7B  |  51.33   | 94.79  |
> |       Cambrian-1*      |   8B  |  55.60   | 22.28  |
> |         SPHINX*        |   7B  |  53.13   |  6.42  |
> |  LLaVA-NEXT-Interleave |   7B  |  57.64   | 16.58  |
> | InternLM-XComposer2-VL |   7B  |  55.85   | 17.94  |
> |     LLaVA-OnVision     |   7B  |  51.77   |  4.90  |
> |      MiniCPM-V2.6      |   8B  |  57.31   | 34.38  |
> |        InternVL2       |   8B  |  59.97   | 30.25  |
> |        LLaVA-1.5       |  13B  |  49.96   | 99.79  |
> |       LLaVA-NeXT       |  34B  |  57.92   | 46.27  |
> |        InternVL2       |  76B  |  68.25   | 55.81  |
>
> From the table, we can see that the recall metric does not show any clear pattern, and it cannot serve as a comparison indicator between different models. However, when combined with accuracy, it can reflect some of the errors made by the models. For example, in the case of LLaVA-1.5, despite its low accuracy, it has a particularly high recall rate, indicating that it classifies almost all samples as abnormal. If you are still interested in such results, we can further test precision and conduct a combined analysis.
>
>
> ### 4. How RAG Improves the Performance
>
> In our paper, RAG is primarily used to retrieve domain knowledge. This domain knowledge provides descriptions of common defects as well as component descriptions of normal objects, enabling the model to better discern the content of the image based on this knowledge, thereby improving its performance. The improvement is not due to the increase in the context length, but rather due to the presence of key information within the context. We provide two examples of RAG-retrieved content in Figures 15 and 16 of Appendix B.
>
> ### 5. Providing the Expert Information Beyond the Visualizations
>
> Thank you for your suggestion. We conducted additional experiments to validate that text can also serve as a form of expert information. Since there is no expert model available to provide information directly from other modalities, we converted the ground truth (GT) into text descriptions of anomaly locations and sizes and provided this to the MLLM. The results are shown in Table 6.
>
> Some of the data from Table 6 are as follows:
>
> | Expert Model | Visualization Type | Anomaly Discrimination | Defect Classification | Defect Localization | Defect Description |
> |--------------|--------------------|------------------------|-----------------------|---------------------|--------------------|
> | GT           | contour            | +10.86                 | +6.45                 | +20.44              | +6.16              |
> | GT           | highlight          | +9.95                  | +1.46                 | +16.59              | +1.21              |
> | GT           | mask               | -1.05                  | +0.69                 | +8.34               | +2.16              |
> | GT           | text               | +24.89                 | +1.36                 | +28.01              | -0.22              |
>
> By comparing with other methods of providing GT information, using text further improves anomaly detection and defect localization capabilities. However, the improvement in perception of categories and appearance is not as significant as that achieved with visual cues.

---

> ### Author Response · Authors · 2024-11-26
> **Rebuttal to Reviewer Comments - 3**
>
> ### 6. Differences between Industrial Anomaly Detection Tasks and General Visual Q&A Tasks
>
> Industrial anomaly detection tasks fundamentally differ from general visual Q&A tasks in several key aspects:
>
> I. **Simplicity and Fixed Context**: Industrial anomaly detection tasks are relatively simple because the scenarios are highly standardized, with little variation between samples. Models are not required to solve complex logical problems, unlike general visual Q&A tasks, which often involve more dynamic, diverse, and abstract visual reasoning.
>
> II. **Focus on Low-level Visual Features**: In contrast to general visual Q&A tasks, which typically focus on object-level features (e.g., identifying objects or their relationships), industrial anomaly detection emphasizes low-level visual characteristics such as texture and structure. These features often lack explicit semantic meaning, making it more challenging to generate accurate and meaningful descriptions.
>
> III. **Availability of Contrasting Samples**: In industrial anomaly detection, obtaining contrasting samples (i.e., normal samples for comparison) is relatively easy and cost-effective. This allows us to provide a template image to the model for comparison, a common practice in industrial inspection. In contrast, general visual Q&A tasks often struggle to define such "standard templates," making them inherently more complex.

---

> > ### Comment · Reviewer_69ie · 2024-11-27
> >
> > I appreciate the authors response to the review. A few quick points:
> > - The authors mentioned the low-level visual feature focus as a unique challenge for this benchmark, which I agree compared with existing visual Q&A problems. The other two differences do not strengthen the claim that industry anomaly detection is a unique challenge.
> > - Appreciate the data on recall; precision, recall and F1 score will together build a more complete picture of model performance in the future.
> > - On multiple-choice format: it is pretty obvious that it is a much easier way to evaluate than open text response, but considering the actual industrial use case in production, it is not realistic to identify the type of anomaly out of four options, unless there is clear justification from experts that there are normally only four failure modes in each scenario. I disagree with author's claim that since the model is able to answer other questions (such as the location of the anomaly), it has a comprehensive understanding of the defect.
> > - I am still not super convinced on the broader value to the ML community, as the benchmark does not shed new light on how solving this industrial specific anomaly detection problem will significantly improve the general MLLM capability gap to real world problems.
> >
> > Therefore, I am maintaining my current score as of now. Happy to hear from authors more about the above points.

---

> > > ### Author Response · Authors · 2024-11-29
> > > **Second Round Responds to Reviewer Comments  - 1**
> > >
> > > Thank you very much for your response, which made us realize some issues in our previous rebuttal. Below, we will focus on addressing the points you raised.
> > >
> > > ### 1. Different Challenges between Industrial Anomaly Detection Tasks and General Visual Q&A Tasks
> > >
> > > I apologize for the misunderstanding in our previous response, where we only reported the differences between Industrial anomaly detection(IAD) and general VQA. The mentioned 'Simplicity and Fixed Context' and 'Availability of Contrasting Samples' might make IAD seem simpler and did not 'strengthen the claim that industry anomaly detection is a unique challenge'. Below, we will further explain how these differences reflect the challenges:
> > >
> > > - **Industrial Scenarios are Simple but High Requirement**: We previously mentioned that industrial images appear to be simpler than natural images. This is because factories achieve better image quality through fixed positions, optimized lighting, and background cropping. However, the high demands of industrial scenarios are reflected in several aspects. In many cases, factories require machines to have higher detection performance than professionals, whereas general visual Q&A tasks usually aim to reach the level of ordinary humans. Specifically, in terms of detection accuracy, the tolerance for missed detections is very low, and in actual projects, zero missed detections might be required, which is a unique challenge compared to natural scenarios. Regarding detection targets, the image resolution is usually large, and the target size is small, with some defects (such as black spots) being only a few pixels, which is rare in general scenarios. Additionally, detection efficiency is often more demanding than in general scenarios because detection needs to keep up with production line efficiency. Our MMAD mainly tests the basic capabilities of anomaly detection, and MLLMs need to reach human-level performance on MMAD before they can be valuable in production environments.
> > >
> > > - **Contrasting Samples as a Unique Prompting Method**: In general VQA tasks, it is rare to provide one or more template images as prompts. This type of prompting increases the context length, raising the detection difficulty, and tests a rarely focused capability of MLLMs - visual comparison. Industrial anomaly detection heavily relies on fine-grained visual comparison rather than semantic-level comparison (some benchmarks have focused on this aspect, such as Seed-Bench2[1] and CompBench[2]). Therefore, the tasks in MMAD are unique.
> > >
> > > Through the discussion during the rebuttal phase, we believe this issue is crucial to understanding MMAD, and we will supplement the results of our discussion in the main text after the discussion concludes.
> > >
> > > ---
> > > [1] Li, Bohao et al. “SEED-Bench-2: Benchmarking Multimodal Large Language Models.” ArXiv abs/2311.17092 (2023): n. pag.
> > >
> > > [2] Kil, Jihyung, et al. "Compbench: A comparative reasoning benchmark for multimodal llms." arXiv preprint arXiv:2407.16837 (2024).

---

> > > ### Author Response · Authors · 2024-11-29
> > > **Second Round Responds to Reviewer Comments - 2**
> > >
> > > ### 2. More Complete Picture of Model Performance with Precision, Recall, and F1 Score
> > >
> > > We appreciate your suggestion and have added a table in the appendix that includes various metrics for anomaly detection (Table 8), a bubble chart with recall and precision as axes (Figure 10), and the corresponding analysis in Appendix A.7. The content is as follows:
> > >
> > > > To comprehensively evaluate the performance of various models in the anomaly discrimination/detection task, we follow the traditional anomaly detection setup, treating the anomaly class as the positive class and the normal class as the negative class. We measured recall, precision, and F1-score. As shown in Table 8, by analyzing the recall and precision metrics, we can identify some reasons for poor accuracy performance in certain models. For instance, both SPHINX and LLaVA-OnVision have recall rates below 10\%, indicating that these models frequently misclassify anomaly samples as normal, leading to a high rate of missed detections. On the other hand, LLaVA-1.5 has a high recall but low precision, suggesting a high rate of false positives. Humans, however, outperform MLLMs across all metrics, with human experts achieving over 94\% and ordinary individuals achieving over 87\%. This disparity is visualized in the bubble chart, as shown in Figure 10, where there is a significant gap between the bubbles representing humans and those representing various MLLMs, and considerable differences among the MLLMs themselves. Additionally, it can be observed that the model AnomalyGPT, which is specifically trained for anomaly detection, performs better than most models but still suffers from a significant false positive issue.
> > >
> > > > Table 8: Performance comparison of different models in anomaly discrimination/detection tasks.
> > >
> > > |          Model         | Scale | Accuracy | Recall | Precision |   F1   |
> > > |:----------------------:|:-----:|:--------:|:------:|:---------:|:------:|
> > > |     Human (expert)     |   -   |  95.24   | 94.25  |   98.89   | 96.43  |
> > > |    Human (ordinary)    |   -   |  86.90   | 87.07  |   94.35   | 89.30  |
> > > |    claude-3.5-sonnet   |   -   |  60.14   | 30.87  |   76.75   | 41.92  |
> > > |    Gemini-1.5-flash    |   -   |  58.58   | 78.63  |   67.41   | 72.40  |
> > > |     Gemini-1.5-pro     |   -   |  68.63   | 45.47  |   86.84   | 57.60  |
> > > |       GPT-4o-mini      |   -   |  64.33   | 65.47  |   73.04   | 68.67  |
> > > |         GPT-4o         |   -   |  68.63   | 67.37  |   75.68   | 71.04  |
> > > |       AnomalyGPT       |   7B  |  65.57   | 82.11  |   74.45   | 76.68  |
> > > |      Qwen-VL-Chat      |   7B  |  53.65   | 43.95  |   65.39   | 47.28  |
> > > |        LLaVA-1.5       |   7B  |  51.33   | 94.79  |   62.72   | 75.32  |
> > > |       Cambrian-1*      |   8B  |  55.60   | 22.28  |   74.10   | 31.85  |
> > > |         SPHINX*        |   7B  |  53.13   |  6.42  |   99.74   | 10.61  |
> > > |  LLaVA-NEXT-Interleave |   7B  |  57.64   | 16.58  |   90.83   | 25.64  |
> > > | InternLM-XComposer2-VL |   7B  |  55.85   | 17.94  |   75.87   | 27.16  |
> > > |     LLaVA-OnVision     |   7B  |  51.77   |  4.90  |   78.19   |  9.10  |
> > > |      MiniCPM-V2.6      |   8B  |  57.31   | 34.38  |   70.98   | 45.31  |
> > > |        InternVL2       |   8B  |  59.97   | 30.25  |   79.22   | 41.23  |
> > > |        LLaVA-1.5       |  13B  |  49.96   | 99.79  |   62.00   | 76.28  |
> > > |       LLaVA-NeXT       |  34B  |  57.92   | 46.27  |   69.98   | 54.44  |
> > > |        InternVL2       |  76B  |  68.25   | 55.81  |   83.52   | 64.40  |
> > >
> > > > **Figure 10: Bubble chart with recall and precision as axes in anomaly discrimination/detection task. (Please refer to the PDF for the chart)**

---

> ### Author Response · Authors · 2024-11-29
> **Second Round Responds to Reviewer Comments - 3**
>
> ### 3. Multiple Choice Question Format
> We appreciate the reviewer's discussion on this issue. Below, we explain our rationale for using four-option multiple-choice questions:
>
> - **Open-text responses may not be the way MLLMs are applied in industrial settings.** MLLMs are likely to be implemented in industrial scenarios through multiple-choice questions due to their greater stability. Consider that with open-text responses, it would be difficult to constrain the model's output range, making it challenging to determine how to process samples. In actual industrial production, Quality Assurance experts typically provide categories and descriptions for high-frequency defects while categorizing long-tail defects under "other."
>
> - **Too many options might prevent accurate evaluation of models.** Four-option multiple-choice questions are the most common format used by humans and are what most MLLMs have been trained on. Having too many options would create longer contexts, making it difficult for some models to properly understand each option. MMAD primarily aims to measure different models' defect perception and judgment capabilities, and we want to avoid any unfairness caused by the question format, hence our choice of the four-option format.
>
> - **The four-option format can also accommodate scenarios with multiple options.** For instance, if there are 16 possible defect categories, we can first divide them into four groups, have the model select the most likely option from each group, and then recombine the selected four options for the model to choose from. Thus, our evaluation method can be extended to more options. In practice, the MMAD dataset includes numerous repeated objects and defects, but each sample has different options, making MMAD's evaluation comprehensive. For example, when questioning about "large broken" defects on a "bottle," the first sample's distractors are "Foggy appearance," "Color fading," and "Scratched surface," while the second sample's distractors are "Smudge," "Discoloration," and "Scratch," with only "Scratch" being repeated.
>
> We will add this discussion to the appendix after the discussion concludes. We also think that open-text responses or additional options also have evaluation value, and we will explore these in future research.
>
> ---
> ### 4. Broader Value to the ML Community
>
> We believe this question is a natural extension of our discussion on "industry anomaly detection is a unique challenge." The unique requirements of IAD mean that solving IAD problems can significantly improve the general MLLM capability gap to real-world problems. The reasons are as follows:
>
> - The high requirements of industrial tasks mean that MLLMs solving IAD problems must possess greater stability.
> - The focus on low-level visual features contributes to understanding real-world problems.
> - The multi-image understanding and image comparison capabilities required in IAD can inspire other real-world domains, such as lesion detection in medical imaging and change detection in remote sensing images.
> ---
> Again, we thank you for providing valuable feedback and taking the time to share your thoughts. In conclusion, we believe this research work has clear value, and we hope you can help make this work visible to the community to contribute to the development of the field. We are committed to addressing any remaining concerns.

---

> > ### Comment · Reviewer_69ie · 2024-11-29
> >
> > I appreciate the authors' response for my comments. I still have reservations on the multiple choice format, as in production setting, users want to know exactly which type of anomaly the part has. The four choice option contains pretty significant bias for defects and it might be hard to detect anomalies that are outside of the "typical categories".
> >
> > However I appreciate authors' efforts on improving the paper to address other concerns, so I am increasing the score to a 6.

---

> > > ### Author Response · Authors · 2024-12-01
> > >
> > > Thank you for the valuable comments and recognition of our work.
> > >
> > > Indeed, our proposed dataset can already test open-ended responses by simply removing the multiple-choice options. Open-ended responses are more challenging as they are without the guidance of options and require not only perception but also expression. However, the current situation shows that even the most powerful model, GPT-4o, performs relatively poorly on multiple-choice questions about the anomaly, showing a significant gap compared to human performance.
> > >
> > > Previous works have tested GPT-4V on several IAD samples using an open-ended response format [1,2], and their conclusions align with our findings: existing MLLMs can only correctly perceive obvious flaws but struggle to find more subtle defects. So, we chose the multiple-choice format to better evaluate those MLLMs and show the performance gap at present. However, we still acknowledge that using various evaluation formats can help prevent test biases, and we will continue to explore potential improvements in the future.
> > >
> > > ---
> > > [1] Yang, Z., Li, L., Lin, K., Wang, J., Lin, C., Liu, Z., & Wang, L. (2023). The Dawn of LMMs: Preliminary Explorations with GPT-4V(ision). ArXiv, abs/2309.17421.
> > >
> > > [2] Cao, Y., Xu, X., Sun, C., Huang, X., & Shen, W. (2023). Towards Generic Anomaly Detection and Understanding: Large-scale Visual-linguistic Model (GPT-4V) Takes the Lead. ArXiv, abs/2311.02782.

---

### Official Review · Reviewer_nkBV · 2024-11-07

**Soundness:** 3
**Presentation:** 3
**Contribution:** 3
**Rating:** 6
**Confidence:** 3

**Summary:**

The authors introduce MMAD, a benchmark specifically designed for anomaly detection in industrial settings to evaluate the capabilities of multimodal large language models (MLLMs). The benchmark encompasses multiple subtasks, wirth 8,336 images and 39,672 questions. Evaluation of current state-of-the-art models demonstrates that performance levels (approximately 70% accuracy) are not yet acceptable by industry standards, highlighting significant gaps in current models’ capabilities. To address these shortcomings, the authors propose the use of Retrieval Augmented Generation for domain-specific knowledge and visual “Expert Agents” to enhance MLLMs’ perception of anomalies. MMAD highlights the current limitations of MLLMs in fine-grained industrial knowledge and provides a benchmark for evaluating future methods in this domain.

**Strengths:**

Comprehensive Benchmark: MMAD is the first benchmark of its kind in industrial anomaly detection, specifically tailored for evaluating MLLMs. It covers a wide range of tasks related to industrial anomaly detection.

Diverse subtasks and Categorization: The benchmark includes numerous subtasks and defect types, ensuring fine-grained evaluations across different aspects of anomaly detection.

The paper is well-written and easy to follow.

**Weaknesses:**

**Lack of Details on Human Supervision:** Given that this is a dataset-centric paper, there is a lack of detailed information about the exact processes used for dataset filtering and the criteria applied during manual verification. More transparency in the data curation process would strengthen the credibility and reproducibility of the benchmark.

**Insufficient Diversity Analysis:** While the dataset is generated from multiple seed datasets and includes diverse subtasks, there is no systematic analysis of the dataset’s diversity (e.g., semantic similarity measures). Furthermore, there is a lack of analysis on the diversity of the questions generated by the LLM. It is unclear whether the questions follow only a few fixed templates or truly represent a wide range of scenarios, which could limit the benchmark’s effectiveness in evaluating models comprehensively.

**Lack of Comprehensive Analysis:** The paper generally lacks in-depth analyses, such as qualitative assessments, error analyses, or ablation studies, which are crucial for evaluating the quality and limitations of both the dataset and the proposed methods. For instance, although the authors note that models perform poorly on defect-related questions, they do not provide detailed insights into the specific types of errors or misconceptions the models exhibit.

**Questions:**

See Weaknesses Above.

---

> ### Author Response · Authors · 2024-11-25
> **Rebuttal to Reviewer Comments - 1**
>
> We appreciate the reviewer’s thoughtful feedback and constructive suggestions, which have helped us improve our paper. Based on your comments, we have made several revisions, detailed as follows:
>
> ### 1. Details on Human Supervision
>
> We have added a detailed description of the human supervision process for dataset filtering in Appendix A.5, which includes both textual explanations and an illustration of the human filtering tool. The text now reads as follows:
>
> *Our textual data is generated by MLLMs, so its accuracy requires human supervision. In our designed process, human supervision is responsible for filtering the final model-generated multiple-choice questions. We first perform preliminary filtering through the program and then enable annotators to conduct an item-by-item review. A total of 26 personnel were involved in the review process, all of whom are researchers in industrial inspection or computer vision and possess a certain level of expertise. However, it should be noted that the industrial inspection field is highly specialized, with significant differences between products. Our annotators' understanding may differ from the professionals' understanding of where the original image data originates. We have developed a tool to enable annotators to filter out problematic multiple-choice questions quickly. As illustrated in Figure 8, we provide the original annotation information through visual and textual means to help annotators accurately identify objects and defects. At the same time, annotators do not need to correct every issue; they simply mark the erroneous questions for exclusion, significantly improving efficiency.*
>
> ### 2. Diversity Analysis of Dataset
>
> We agree that a diversity analysis of the dataset is important. While we have already analyzed the diversity of image categories and defect types in the original manuscript, we have expanded the analysis to include semantic diversity in Appendix A.6, as suggested. This section now includes both visual and textual information, with word clouds showing the distribution of question and option terms. It is important to note that in the context of industrial inspection, questions are typically less diverse than in natural scene tasks, as the scenarios are more fixed, and the questions generally revolve around defect detection, categories, and appearance. Therefore, while the diversity of questions in our MMAD benchmark is relatively low, the diversity of options is higher.
>
> The added text reads as follows:
>
> *To systematically analyze the diversity of the dataset, we separately calculated the frequency of phrases appearing in questions and options, presenting them in the form of word clouds in Figure 9. In the question text, the word “defect” appeared most frequently, followed by “object,” reflecting that our benchmark is constructed for industrial inspection scenarios, focusing on seven sub-tasks. In addition to common words such as “image,” “type,” and “appearance,” the questions also include diverse expressions with lower frequencies. For example, expressions indicating position such as “relative position,” “arrangement,” and “defect located.” In the text of the options, “Yes” and “No” are the standard answers for anomaly discrimination questions, thus appearing most frequently. Beyond these two words, the diversity of options is more evident, including specific descriptions of various positions or expressions of different types of defects.*

---

> ### Author Response · Authors · 2024-11-25
> **Rebuttal to Reviewer Comments - 2**
>
> ### 3. Comprehensive Analysis
>
> We would like to clarify that we have already included some in-depth analyses in the original manuscript. For instance, Section 4.3 contains analyses on the utilization of template images in MLLMs, the impact of model size, and the effect of the number of template images. Section A.2 discusses in-context learning. After reviewing your comments, we have added more extensive analyses to the appendix. Specifically, we have updated three qualitative analyses in Appendix B, included an analysis of the impact of the Chain of Thought in Section A.3, and added an ablation study of the visual components in Section A.4.
>
> The added text reads as follows:
>
> *A.3 ANALYSIS OF CHAIN OF THOUGHT*
>
> *Chain of Thought (CoT) is a commonly used method to enhance the logical reasoning abilities of MLLMs (Chu et al., 2024). To investigate whether current MLLMs lack reasoning when addressing MMAD problems, we introduced a straightforward CoT approach. The process involves three steps: first, the model identifies objects in the image; second, it compares the differences between the template image and the query image; and finally, it determines whether the identified difference constitutes a defect in the object. We incorporated a set of rules and adjusted the instructions accordingly, dividing the CoT responses into two stages. As shown in the Table 6, InternVL2-76B, the best-performing open-source MLLM we tested, achieved a 1.5\% improvement in CoT-based performance. However, its anomaly detection accuracy, which is the most critical metric, showed no improvement. On the other hand, InternVL2-40B experienced a performance decline after introducing CoT, potentially due to insufficient stability in the language model's reasoning capabilities.*
>
> Table 6: Performance comparison of different MLLMs with and without CoT.
> | Model             | CoT        | Anomaly Discrimination | Defect Classification | Defect Localization | Defect Description | Defect Analysis | Object Classification | Object Analysis | Average |
> |-------------------|------------|------------------------|-----------------------|---------------------|--------------------|-----------------|-----------------------|-----------------|---------|
> |                   | -          | 64.45                  | 50.57                 | 53.42               | 66.17              | 79.56           | 90.65                 | 82.36           | 69.59   |
> | **InternVL2-40B** | ✔️         | 59.42                  | 46.10                 | 51.92               | 57.66              | 79.80           | 80.55                 | 85.29           | 65.82   |
> |                   | -          | 68.25                  | 54.22                 | 56.66               | 66.30              | 80.47           | 86.40                 | 82.92           | 70.75   |
> | **InternVL2-76B** | ✔️         | 68.18                  | 54.61                 | 58.64               | 68.89              | 79.95           | 90.51                 | 85.25           | 72.29   |
>
> *A.4 ANALYSIS OF VISION DISABLE*
>
> *Some studies (Tong et al., 2024a; Chen et al., 2024a) have proposed that determining whether a
> benchmark requires visual input to be solved has been a persistent challenge in vision-language
> research. To validate MMAD, we masked the visual components and compared the performance
> of MLLMs with and without visual input. The results, as shown in the Table 7, indicate that most
> subtasks in MMAD are highly dependent on the visual components. Performance significantly
> decreases when the visual input is removed.*

---

> ### Author Response · Authors · 2024-11-25
> **Rebuttal to Reviewer Comments - 3**
>
> Table 7: Performance comparison of different MLLMs with and without vision.
> | Model            | Vision  | Anomaly Discrimination | Defect Classification | Defect Localization | Defect Description | Defect Analysis | Object Classification | Object Analysis | Average |
> |------------------|---------|------------------------|-----------------------|---------------------|--------------------|-----------------|------------------------|-----------------|---------|
> |                  | Enable  | 59.97                  | 43.85                 | 47.91               | 57.60              | 78.10            | 74.18                  | 80.37           | 63.14   |
> | **InternVL2-8B** | Disable | 49.71                  | 32.28                 | 41.38               | 52.20              | 75.34            | 41.94                  | 57.13           | 50.00   |
> |                  | Enable  | 63.84                  | 51.58                 | 52.94               | 66.43              | 79.75            | 90.76                  | 82.41           | 69.67   |
> | **InternVL2-40B**| Disable | 49.61                  | 24.43                 | 36.35               | 46.39              | 68.05            | 37.29                  | 55.75           | 45.41   |
> |                  | Enable  | 68.25                  | 54.22                 | 56.66               | 66.30              | 80.47            | 86.40                  | 82.92           | 70.75   |
> | **InternVL2-76B**| Disable | 49.93                  | 25.92                 | 28.54               | 43.88              | 70.41            | 35.89                  | 57.53           | 44.59   |
>
> *Figure 10: A case of a question-and-answer result in qualitative analysis. Since MMAD is test-
> ing with multiple-choice questions, we cannot directly analyze the answers. Instead, we use the
> chain-of-thought approach to encourage the model to provide its own analysis. In this example, the
> InternVL-40B model did not notice the damage to the cable, yet it still chose the correct answer
> from the options. In comparison, GPT-4o can output a proper analysis and letter. Subsequent issues
> such as anomaly classification, localization, and description can further test whether the model’s cor-
> rect answers are coincidental. Therefore, a comprehensive evaluation, rather than simple anomaly
> detection, is essential.*
>
> *Figure 11: A case of a question-and-answer result in qualitative analysis. A human can quickly
> notice the defect in the query image, while the model focuses on the number of components. This
> illusion tends to occur when the defect is minor, and the object’s composition is complex.*
>
> *Figure 12: A case of a question-and-answer result in qualitative analysis. For object-related questions, GPT-4o possesses extensive knowledge and can analyze object information based on subtle clues. However, most humans do not have such comprehensive knowledge. For instance, non-specialists may not recognize BAT+ and BAT-, let alone the TC4056A IC chip.*

---

> ### Comment · Reviewer_nkBV · 2024-11-26
>
> I thank the authors for providing comprehensive evaluations and clarifying several points. I have increased my soundness score and contribution score accordingly!

---

> ### Author Response · Authors · 2024-11-29
>
> Dear Reviewer nkBV,
>
> Thank you for your swift response and the revised evaluation of our paper. We sincerely appreciate the time and effort you have dedicated to reassessing our work. Your initial feedback was instrumental in enhancing our paper.
>
> We believe that an improved rating could help this pioneering work gain the visibility it deserves within the community. We kindly request that you consider raising the rating of our paper. Should there be any additional improvements or unresolved issues that might impact the review, we would be grateful for your suggestions. We are committed to resolving any remaining concerns.
>
> Thank you once again for your support.
>
> Best regards,
> The Authors

---

### Meta-Review · Area_Chair_rRvY · 2024-12-22

**Metareview:**

The paper proposed a new benchmark for vision language models on the task of detecting industrial anomalies and defects. Reviewers generally find the contribution of the paper valid and valuable as the application of LLMs to industrial settings is important. The writing quality is good. The main weakness left unaddressed after the rebuttal is the novelty of the benchmark (compared to other image QA benchmarks) and its relevance to the ICLR community. Considering this benchmark has its own unique characteristics (e.g., defects detection requires fine-grained vision comparisons) and the broad interests in LLM applications in the industry, the AC is inclined to accept this paper.

The AC **strongly advises the authors to remove the word "first-ever"** from the title of this paper. Using this word is simply unprofessional in academic papers - every paper can claim their contribution as "first-ever" and this word is largely redundant and unnecessary.

**Additional Comments On Reviewer Discussion:**

Reviewers and authors had a few rounds of conversations during the rebuttal period, and most concerns have been addressed. All reviewers are positive about the paper after the rebuttal.

---

### Decision · Program_Chairs · 2025-01-22

Accept (Poster)